# Structural basis for assembly of the CBF3 kinetochore complex

Vera Leber[1], Andrea Nans[2] & Martin R Singleton[1],*

## Abstract

Eukaryotic chromosomes contain a specialised region known as the centromere, which forms the platform for kinetochore assembly and microtubule attachment. The centromere is distinguished by the presence of nucleosomes containing the histone H3 variant, CENP-A. In budding yeast, centromere establishment begins with the recognition of a specific DNA sequence by the CBF3 complex. This in turn facilitates CENP-A$^{Cse4}$ nucleosome deposition and kinetochore assembly. Here, we describe a 3.6 Å single-particle cryo-EM reconstruction of the core CBF3 complex, incorporating the sequence-specific DNA-binding protein Cep3 together with regulatory subunits Ctf13 and Skp1. This provides the first structural data on Ctf13, defining it as an F-box protein of the leucine-rich-repeat family, and demonstrates how a novel F-box-mediated interaction between Ctf13 and Skp1 is responsible for initial assembly of the CBF3 complex.

**Keywords** CBF3; centromere; F-box protein; kinetochore; single-particle cryo-EM

**Subject Categories** Cell Cycle; Chromatin, Epigenetics, Genomics & Functional Genomics; Structural Biology

The EMBO Journal (2018) 37: 269–281

## Introduction

The connection between the mitotic spindle and chromosomes in eukaryotes is formed by the kinetochore, a large, dynamic, regulated multi-protein complex (Westermann *et al*, 2007; Santaguida & Musacchio, 2009). The kinetochore assembles on a single, specific chromosomal locus known as the centromere. This is distinguished by the presence of a specialised centromeric nucleosome known as CENP-A (Cse4 in budding yeast; McKinley & Cheeseman, 2016). The unique structural properties of CENP-A drive nucleation and assembly of the rest of the kinetochore and ultimately formation of spindle-chromosomal attachments (Sekulic *et al*, 2010). How CENP-A nucleosomes are directed to incorporate into chromatin at the appropriate sites is a subject of ongoing interest. In budding yeast, the so-called point centromere comprises ~125 bp of DNA which is necessary and sufficient to allow kinetochore establishment (Fitzgerald-Hayes *et al*, 1982). This short sequence, sub-divided into elements CDEI, CDEII and CDEIII, is capable of binding the centromere-binding factors Cbf1 and CBF3, as well as a centromeric nucleosome (Clarke & Carbon, 1980; Lechner & Carbon, 1991; Meluh *et al*, 1998; Cole *et al*, 2011). How all these factors assemble on such a short sequence is a matter of some debate. While the Cbf1 protein is dispensable for cell viability (Cai & Davis, 1990), the CBF3 complex is essential and forms the keystone for subsequent kinetochore assembly. The ~445-kDa CBF3 complex consists of four proteins: Ndc10, Cep3, Ctf13 and Skp1 present in a 2:2:1:1 stoichiometry (Lechner & Carbon, 1991; Doheny *et al*, 1993; Lechner, 1994; Strunnikov *et al*, 1995; Stemmann & Lechner, 1996; Russell *et al*, 1999; Espelin *et al*, 1997). Previous studies have suggested that Ndc10 and Cep3 constitute the DNA-binding activities of the complex. Both Ndc10 and Cep3 are capable of directly binding DNA (Espelin *et al*, 1997; Purvis & Singleton, 2008; Cho & Harrison, 2011; Perriches & Singleton, 2012); in the case of Cep3, sequence specificity for the CDEIII element is provided by zinc-containing Gal4-type DNA-binding domains (Strunnikov *et al*, 1995; Espelin *et al*, 1997). The function of Ctf13 and Skp1 has remained obscure. Early studies of CBF3 showed that its assembly is highly regulated in the cell, with multiple phosphorylation/ubiquitination activities required as well as the involvement of the Hsp90-Sgt1 protein chaperone pathway (Stemmann & Lechner, 1996; Kaplan *et al*, 1997; Kitagawa *et al*, 1999; Russell *et al*, 1999; Stemmann *et al*, 2002; Bansal *et al*, 2004; Lingelbach & Kaplan, 2004; Rodrigo-Brenni *et al*, 2004). The principal target of these activities appears to be Ctf13 and Skp1. Exactly how these pathways contribute to the assembly of mature, centromere-binding proficient CBF3 is still not entirely clear. Crystal structures have been determined of the truncated Cep3 and Ndc10 proteins (Bellizzi *et al*, 2007; Purvis & Singleton, 2008; Cho & Harrison, 2011; Perriches & Singleton, 2012) as well as full-length Skp1 (Schulman *et al*, 2000; Orlicky *et al*, 2003). However, we are missing structural data for Ctf13 and crucially, the intact CBF3 complex.

Here, we present the structure of the 220-kDa Cep3-Ctf13-Skp1 (henceforth "core") complex determined by cryo-electron microscopy at an overall resolution of 3.6 Å. As well as providing the first structural data on the Ctf13 protein, we can rationalise the assembly of CBF3 and better explain the unusual DNA-binding properties of

1 Structural Biology of Chromosome Segregation Laboratory, The Francis Crick Institute, London, UK
2 Structural Biology of Cells and Viruses Laboratory, The Francis Crick Institute, London, UK
*Corresponding author. Tel: +44 203 796 2034; E-mail: martin.singleton@crick.ac.uk

   The EMBO Journal   Vol 37 | No 2 | 2018   **269**

Cep3. We also uncover a new mode of Skp1/F-box interaction, which may have more general implications, and discuss possible models for the formation of the complete complex.

## Results

### Expression of the CBF3 complex

Although recombinant expression of the isolated Ndc10, Cep3 and Skp1 proteins is readily achieved, it has proven extremely difficult to express and purify active Ctf13, possibly because of post-translational modifications and chaperone activity required to stabilise the protein and assist complex formation. To circumvent this problem, we co-expressed all four full-length proteins as well as the known assembly factor Sgt1 in budding yeast under the control of an inducible GAL promoter. This enabled purification of intact CBF3 complex, with all four subunits present in the predicted Ndc10(2):Cep3(2):Ctf13(1):Skp1(1) stoichiometry as judged by band intensities. However, we found that Ndc10 can readily dissociate from the Cep3-Ctf13-Skp1 core complex, leading to the purification of both the full and the core complex (Fig 1A). Although size-exclusion profiles for both complexes looked promising (Fig 1B), negative stain electron microscopy showed that only the core complex forms monodisperse, distinct particles suitable for structural analysis (Fig EV1A). The full complex on the other hand is comprised of core particles and additional diffuse density, suggesting that a large part of Ndc10 is unstructured (Fig EV1B). It has been shown that the structured N-terminal domain (NTD) of Ndc10 can bind to the core complex (Cho & Harrison, 2011). Therefore, we tested whether a complex comprising the core and only the NTD of Ndc10 (residues 1–551) would be suitable for structural analysis. Although the co-expression was successful (Fig EV1C), the interaction between the Ndc10 NTD and the core complex seems to be less stable than with full-length Ndc10 (Fig EV1D) and negative stain analysis showed no improvement of particles (Fig EV1E). Therefore, further structural analysis was only carried out with the core complex. We also tested whether the C-terminal domain (CTD, residues 552–956) of Ndc10 can still interact with the core by co-expression. However, co-expression was not successful, suggesting that the NTD is required but not sufficient for interaction (Fig EV1C).

### Structure determination

Images of frozen-hydrated core complex were recorded with a Titan Krios and K2 camera (Fig 1C). Multiple rounds of 2D classification and a single round of 3D classification yielded class averages with clear secondary structure visible (Fig 1D). Although there was some particle orientation bias apparent and resultant resolution anisotropy in the final maps (Tan et al, 2017; Appendix Fig S1A and D), we were able to obtain a final reconstruction with an overall resolution of 3.6 Å (Appendix Table S1, Appendix Fig S1B and C). 3D classification of structural heterogeneity using standard techniques (Scheres, 2010) did not reveal any significant conformational variability.

The final map obtained allowed us to dock the known crystal structures of Cep3Δ (N-terminal Gal4-domain deleted) and Skp1 (Fig 2A–E). The electron density was well defined for the majority

of the Cep3 and Skp1 proteins, but somewhat more variable for Ctf13 (Appendix Fig S2A–C). The Cep3 structure closely matches the earlier crystal structures (Bellizzi et al, 2007; Purvis & Singleton, 2008), with some additional density being apparent for a previously unresolved loop (Appendix Fig S2D), as well as one of the Gal4 DNA-binding domains (see below, Fig 2D). The extra protein features were built into the structures, sequence assigned and the Cep3 and Skp1 models refined using real-space refinement. Refinement statistics are presented in Appendix Table S2.

Together, Ctf13 and Skp1 form a tightly associated dimer which sits on the outside lower edge of the Cep3 crescent (Fig 2A), forming a central channel of about 30 Å diameter in the overall shape (Fig 2B). The majority of the Skp1 crystal structure fits directly into the EM reconstruction; however, there is some movement occurring in the C-terminal helices to accommodate binding of the Ctf13 F-box. In addition, an extended loop between residues 105 and 112, missing in all current crystal structures, is resolved in the EM map. The significance of these features is discussed further below.

### Ctf13 is an F-box protein with leucine-rich repeats

Our structure provides the first insight into the shape of Ctf13. The density clearly shows that the N-terminal F-box forms part of a small helical domain, which is followed by seven leucine-rich repeats (LRRs; Fig 3A–C), placing Ctf13 in the sub-class of FBXLs (F-box proteins containing LRRs; Kipreos & Pagano, 2000). A helical insertion in the LRRs wraps around the end of the structure distal to the F-box domain (Fig 3C).

For unknown reasons, the quality of the density map in Ctf13 is highly variable. The main structural domains can be clearly defined, and some sequence assigned at the very N-terminus in the F-box and within the LRR helical insertion and C-terminus. However, the connectivity for some of the N-terminal helical domain is unclear, and density for the external linking loops of the LRRs is broken and extremely noisy, making reliable secondary structure or sequence assignment difficult (Appendix Fig S2B). For these reasons, we have not attempted to build a full atomic model for the protein but rather generated a simple poly-alanine model that highlights the general features of the protein (Figs 3C and EV2A and B) and restricted our analysis to features that can be unambiguously determined.

### Skp1/F-box interaction

The ubiquitous cell cycle protein Skp1 is a component of the Skp1–Cullin–F-box (SCF) family of ubiquitin ligases and links a substrate-binding F-box protein to a Cullin (Bai et al, 1996; Skowyra et al, 1997). The interaction between Skp1 and the F-box has been widely studied and is mediated by the three C-terminal helices of Skp1, which wrap around a compact three-helix bundle from the N-terminus of the F-box protein (Schulman et al, 2000). Although the same structural elements are present in the CBF3 complex, we found an interesting difference in the Skp1/F-box interaction compared to current known structures. The same C-terminal helices (helices α6, α7, α8) of Skp1 are involved in the interface; however, they lie in a more closed conformation with the extended loop between the α7 and α8 helices folded and shortened (Fig 4A–C). The F-box of Ctf13 hooks underneath the α6 helix of Skp1 in a similar fashion to other F-box proteins, but the α2 and α3 helices of the

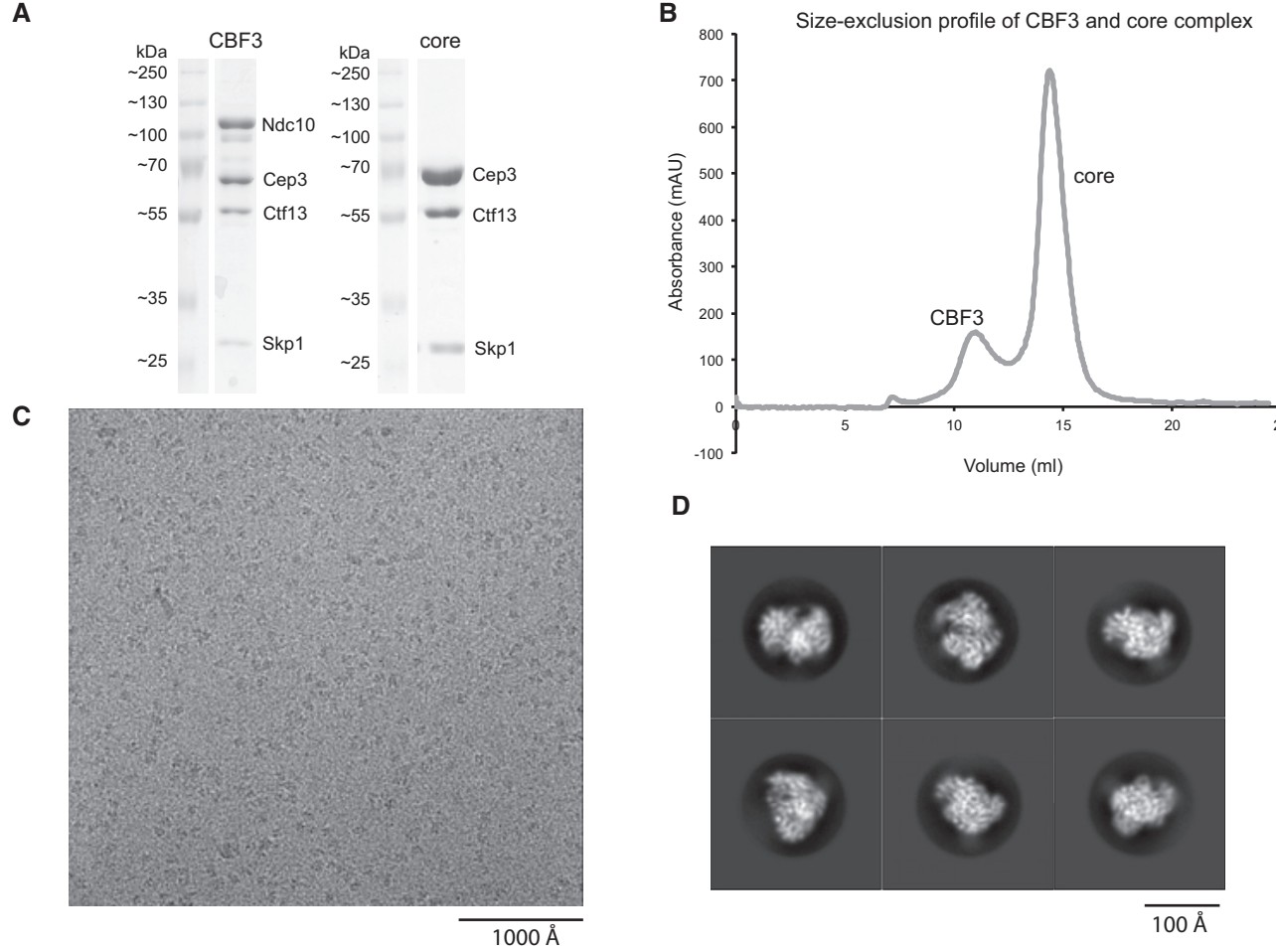

**Figure 1.  CBF3 biochemistry and structural analysis.**

A   SDS–PAGE/Coomassie blue stain of the purified CBF3 full and core complex after size-exclusion chromatography.
B   Size-exclusion profile showing the full CBF3 complex (CBF3) and the core complex (core).
C   Example micrograph of CBF3 core complex embedded in a thin layer of vitreous ice and recorded at a defocus of −3 μm.
D   2D-classes of different representative views, showing defined secondary structures.

Ctf13 F-box are in a substantially different orientation (Fig 4C). There is also an extended interface between the F-box of Ctf13 and the α8 helix of Skp1 (Fig 4A–C), as well as an extended loop of Skp1 (residues 105–112, Fig 4D). Interestingly, this loop, conserved between yeast and humans (Fig 4E), is found to be disordered in crystal structures. In our EM reconstruction however, there is clear density for this loop, showing that it interacts with both the LRRs and the F-box of Ctf13.

**Interactions between Ctf13, Skp1 and Cep3**

Our structure provides a first understanding of the interactions between Cep3, Ctf13 and Skp1. As described, Ctf13 and Skp1 form a heterodimer, which sits on the outside edge of one Cep3 monomer. Interestingly, the heterodimer also stabilises the Gal4-domain of the same monomer, allowing us to dock a known homologous crystal structure (King *et al*, 1999). The quality of the density is sufficient to visualise the $Zn_2Cys_6$ cluster, assign the correct amino acid

sequence and the build the likely path of the loop connecting the domain to the rest of the Cep3 protein (Fig 5A and B). The second Gal4-domain is not resolved in the map, which is unsurprising given its connection to the rest of the protein through a flexible linker. The impact of the stabilisation of the one Gal4-domain on DNA binding is discussed further below.

To answer the question of why only one Ctf13/Skp1 heterodimer binds to a Cep3 homodimer, we superimposed a potential second Ctf13/Skp1 heterodimer on the other Cep3 monomer. This shows a steric clash between the proteins, explaining the Cep3(2):Ctf13(1):Skp1(1) stoichiometry of the complex (Fig EV3A and B). Size-exclusion chromatography coupled with multi-angle light scattering (SEC-MALS) confirmed that solution mass of the core matched this stoichiometry and the EM structure (Fig EV3C).

Previous biochemical studies have shown that Ctf13 interacts with all other subunits and therefore acts as a central core of CBF3 (Russell *et al*, 1999). Our structure confirms that it indeed contacts both Cep3 and Skp1 (Fig 6A and B). Whereas the interaction with Skp1 is mainly

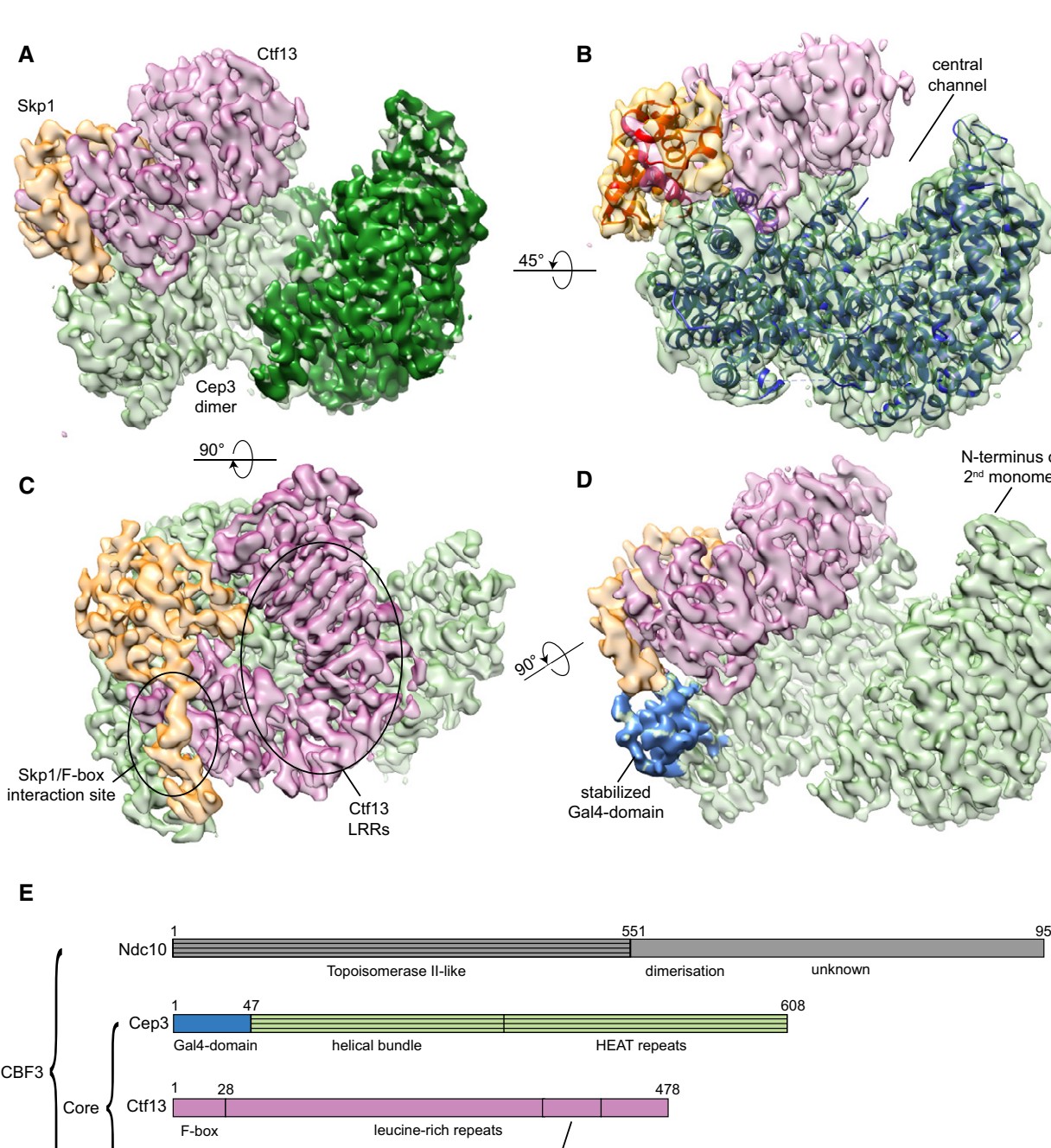

**Figure 2. Cryo-EM reconstruction of the CBF3 core complex.**

Refined and sharpened cryo-EM density maps of CBF3 core complex in different orientations with colour-coded subunits (Cep3 homodimer—green; Ctf13—pink; Skp1—orange).

A   Orientation showing the Ctf13/Skp1 heterodimer sitting on the crescent of one Cep3 monomer, while the other monomer remains unbound.

B   Side view showing the central channel in the overall shape and the fitted crystal structures of Skp1 (PDB ID: 1nex, red) and Cep3Δ (PDB ID: 2veq, blue).

C   Top view showing the Ctf13/Skp1 heterodimer; highlighted are the leucine-rich repeats (LRRs) of Ctf13 and the interaction site of Skp1 with the Ctf13 F-box.

D   Side view, in which the Gal4-domain, adjacent to the Ctf13/Skp1 heterodimer is highlighted in blue. Only this Gal4-domain is stabilised, and the other Gal4-domain remains flexible and is therefore not resolved in the cryo-EM density map. The approximate location of this domain at the N-terminus of the Cep3 monomer is indicated.

E   Schematic diagram of the four CBF3 subunits, colour coded as above.

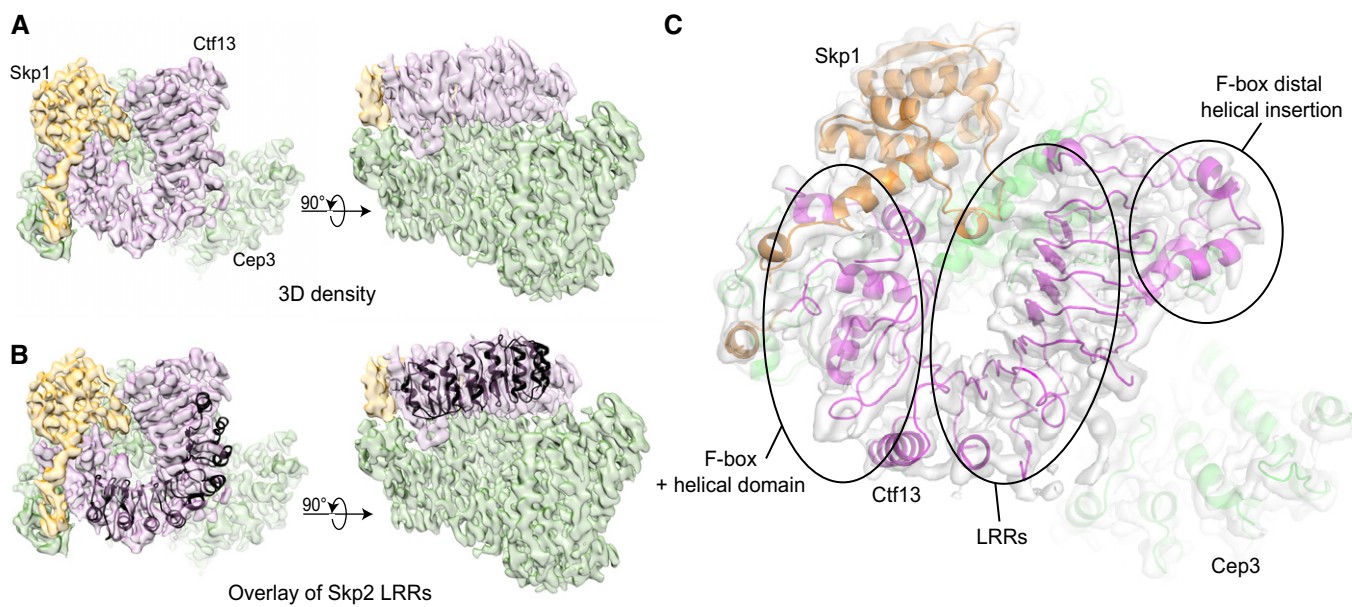

**Figure 3.  Structural insights into Ctf13.**

A   Cryo-EM density maps of the core complex in top (left panel) and side view (right panel) with colour-coded subunits (Skp1—orange, Ctf13—pink, Cep3—green).

B   To visualise the typical arc shape of LRRs, we superimposed the Skp2 LRRs (PDB ID: 1fqv), shown in black on the reconstruction.

C   Detailed top view showing the cryo-EM density in light grey and fitted structures of Skp1 (orange), Cep3 (green) and the model of Ctf13 (pink). Highlighted are the overall structural features of Ctf13: the F-box with the adjacent helical domain, the LRRs and the F-box-distal helical domain, which is inserted in the LRRs.

through its F-box domain and some contacts with the final LRR and the LRR insertion, the interface to Cep3 is much broader and includes contacts made with the F-box and helical domain, the LRRs and the F-box distal helical insertion (Fig EV2B). These results conform very well with earlier biochemical interaction data conducted using *in vitro* translated Ctf13, showing there are two important regions for the interaction between Ctf13 and Skp1: the N-terminal F-box and a C-terminal region. For the interaction between Ctf13 and Cep3, however, most constructs show some binding to Cep3, indicating an extended interface between these proteins (Russell *et al*, 1999).

Our structure shows that Skp1 not only makes contacts with Ctf13, but also Cep3 (Fig 6C). Interestingly, the Skp1 surface involved is also responsible for binding to Cul1 in the SCF ligase context (Fig EV4A–D), highlighting additional structural similarity to the SCF ligase. To test these interactions, we made mutations in Skp1 already shown to inhibit Cul1 binding and tested for complex formation by pull-down assays (Zheng *et al*, 2002). Wild-type or mutant Skp1 (N139K, Y140K) was overexpressed with the other three CBF3 subunits. For detection, all but Ctf13 were triple HA-tagged and Cep3 carried an additional C-terminal double-Strep tag. Lysates were run through a StrepTactin column and eluted with biotin. Eluates were visualised with SDS–PAGE and Coomassie stain, as well as Western blot against the HA-tag. Only wild-type HA-tagged Skp1 could be detected in the eluate, whereas mutant Skp1 was not able to form a complex with the other proteins and was effectively replaced by endogenous Skp1 (Fig 6D). Both wild-type and mutant Skp1 were expressed, however, as they could be detected by Western blot in the lysate in approximately equal amounts. These results confirmed that the same residues of Skp1 (Y139, N140) are essential for CBF3 complex formation, as well as SCF ligase formation.

## DNA binding by Cep3

Cep3 provides sequence-specific DNA binding to the CBF3 complex, utilising a Gal4-like $Zn_2Cys_6$ cluster (Espelin *et al*, 1997; Purvis & Singleton, 2008). The canonical recognition sequence for this DNA-binding motif is a CCG/GGC triplet, and all known Gal4-type transcription factors bind as a dimer to a site containing two copies of this sequence, either as a direct or inverted repeat (Marmorstein *et al*, 1992; Liang *et al*, 1996; Schjerling & Holmberg, 1996). Interestingly, this triplet is only present as a single copy in the CDEIII element, raising the question of how the symmetrical Cep3 dimer binds a single asymmetric site. The structure provides an answer to this question. As previously described, one of the Cep3 Gal4 domains is stabilised by interactions with both Ctf13 and the F-box binding segment of Skp1 (Fig 5A), with clear density visible at the tip of one Cep3 monomer, close to the N-terminal of the crystal structures. The map is of sufficient quality to allow unambiguous fitting and modelling of the atomic structure of a Gal4 domain. These domains have pseudo-dyad symmetry, which in principle would allow them to be fitted in one of two orientations. However, close inspection of the reconstruction shows that one orientation is far more consistent with the observed density. By superimposing the crystal structure of the homologous Hap1 domain (King *et al*, 1999) bound to DNA on the Cep3 Gal4 domain, we can trace the likely path of the DNA. This modelling clearly shows a severe clash between the DNA and the LRRs of Ctf13, indicating that this domain is likely incapable of DNA binding once the complex has formed (Fig EV5A and B). In effect, the binding of the Skp1/Ctf13 heterodimer to Cep3 locks the domain in an inactive conformation. Therefore, the second free Gal4 domain on the other Cep3 monomer must be responsible for binding the CCG site in CDEIII.

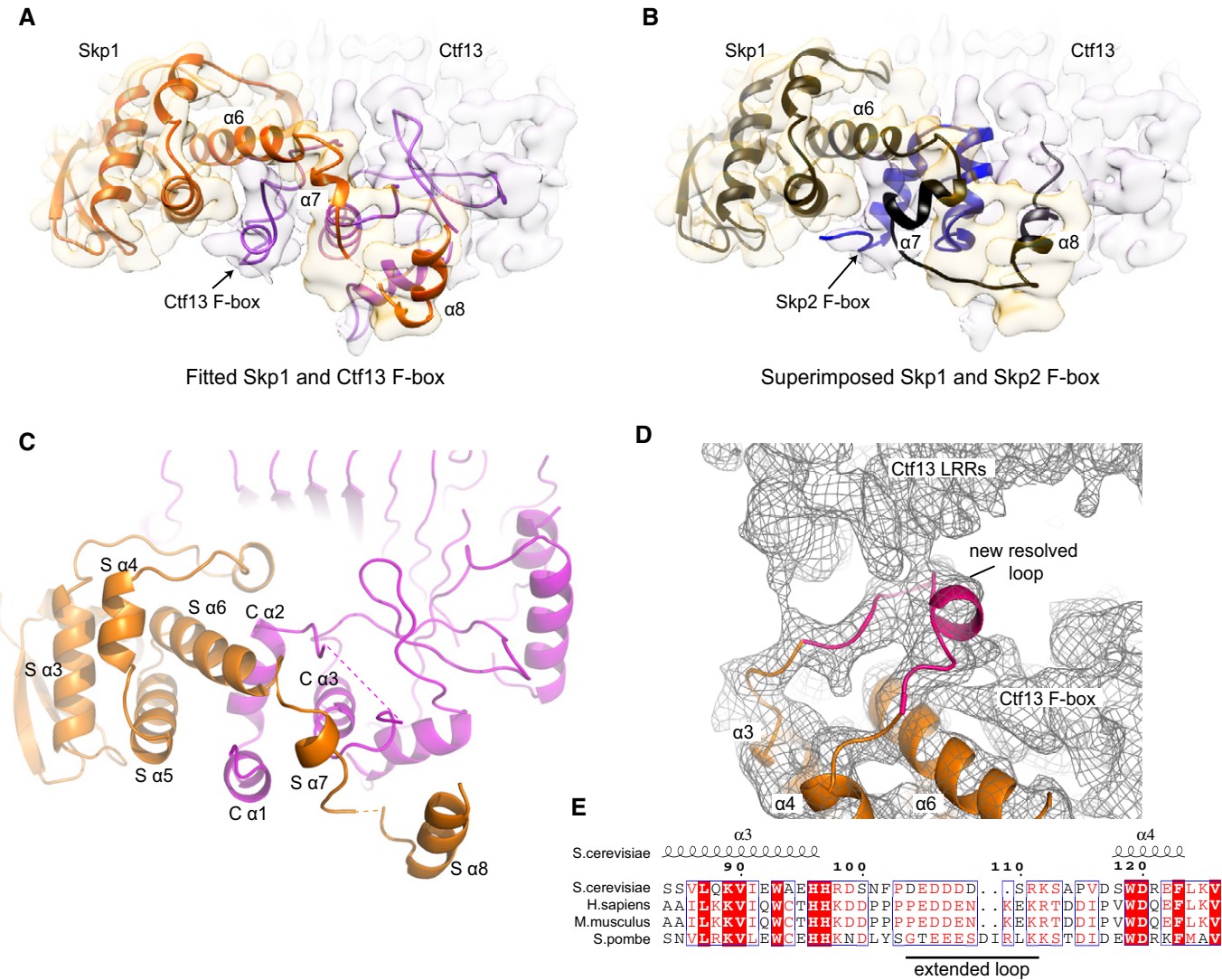

**Figure 4. Skp1 binds the Ctf13 F-box in a novel manner.**

A Detailed view of Skp1/F-box interaction site with cryo-EM density map showing Skp1 in yellow and the Ctf13 in light-pink, as well as the fitted models.

B Skp1 and Skp2 F-box (PDB ID: 1fqv) fitted to density shows differences between crystal structures and our reconstruction. Helices α6, α7 and α8 of Skp1 are labelled.

C Detailed view of the Skp1/F-box interaction shown as ribbon diagram (Skp1—orange, Ctf13—pink), with helices of both Skp1 (S α3–8) and Ctf13 (C α1–3) labelled.

D Detailed view of the structured Skp1 loop (residues 105–112; highlighted in red) and its interaction with the F-box and the LRRs of Ctf13. Helices α3, α4 and α6 of Skp1 are labelled.

E Sequence alignment of Skp1 homologues focused at the loop region shown in (D). Conserved residues are highlighted in red font, identical residues in red fill.

Although some studies have suggested that CBF3 is only capable of binding centromeric DNA with all four subunits present (Espelin *et al*, 1997), it has been shown that Cep3 and Ndc10 alone have DNA-binding ability (Purvis & Singleton, 2008; Cho & Harrison, 2011). We therefore tested whether the core complex is capable of binding to the centromeric DNA. We conducted electrophoretic mobility shift assays, showing that the core complex is indeed capable of centromeric DNA binding (Fig 7A). Interestingly, this binding is dependent on pre-treatment with λ-protein phosphatase (λ-phosphatase) suggesting that there is a phosphorylated, inactive and a dephosphorylated, active state of the complex. To establish whether the Gal4 domains of Cep3 are responsible for the observed DNA-binding ability, we compared the wild-type core complex with one containing a truncated Cep3 protein lacking the N-terminal Gal4 domains (core ΔGal4). As expected, this construct did not show any DNA-binding capability with or without λ-phosphatase treatment. To identify which subunit is dephosphorylated by λ-phosphatase and therefore may be responsible for the differences in DNA binding, we analysed the phosphorylation status of the constituent proteins using SDS–PAGE with the Phos-Tag reagent. This reagent causes an exaggerated gel shift of phosphorylated proteins. After treatment with λ-phosphatase for specified times, the samples were separated on the gels. As shown in Fig 7B, we could observe a shift of Skp1, but not Cep3 and Ctf13. Intact mass spectrometry of Skp1 showed the presence of four phosphorylations (Appendix Fig S4). These results conform very well with previous mass spectrometry

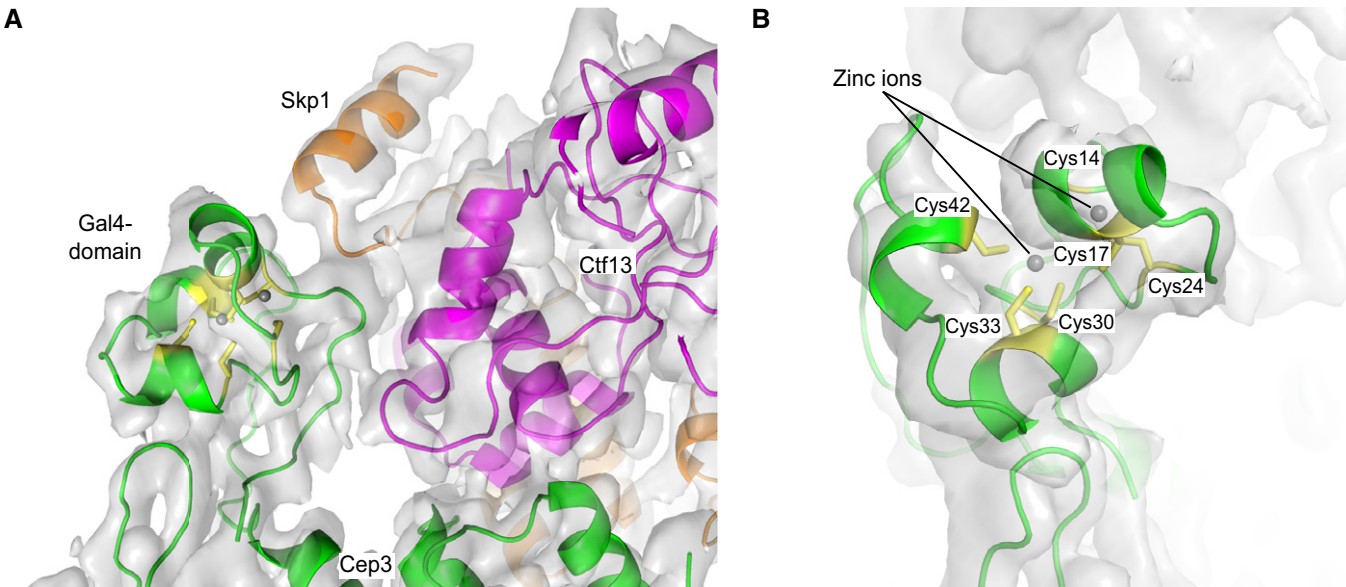

**Figure 5.   The stabilised Gal4-domain.**

A    Detailed view of the stabilised Gal4-domain and its interaction with both Skp1 and Ctf13. The cryo-EM density is shown in grey and the fitted structures of Skp1 in orange, Ctf13 in pink and Cep3 in green. The six cysteines of the Gal4-domain are highlighted in yellow, and the two zinc ions are shown as grey spheres.

B    Top view of the stabilised Gal4-domain, showing the density for the $Zn_2Cys_6$ cluster.

results which showed multiple sites located on a loop between residues 37 and 64 (Stemmann *et al*, 2002). To test whether this loop is responsible for the phosphorylation-dependent difference in DNA binding, we expressed a construct lacking this loop (Skp1Δ) (Connelly & Hieter, 1996). Core complex containing Skp1Δ could bind to the centromeric DNA even without λ-phosphatase treatment (Fig 7C), suggesting that the phosphorylation status of this loop influences the ability of the core complex to bind to DNA. This solvent-exposed loop is probably flexible, as we do not observe it in our density map. It lies near the N-terminus of Skp1, distal to the F-box, and faces away from Cep3 (Fig EV5C). It should be noted that we obtained the core structure without pre-treatment with λ-phosphatase; therefore, the presented structure likely represents an inactive state of the core complex. To test the possibility that this inactivation compromises the interaction with Ndc10, we co-expressed and purified CBF3 Skp1Δ, mimicking the dephosphorylated state. Both core wild type and Skp1Δ were able to interact with Ndc10, as shown by pull-down. Interestingly, more Ndc10 appeared to be bound to the Skp1Δ-containing complex, suggesting an enhanced interaction. However, size-exclusion chromatography and subsequent negative stain analysis showed no difference in the behaviour of the full complex and the Skp1Δ-containing core appeared indistinguishable from the wild type (Appendix Fig S3A–D).

To confirm that the DNA binding we see is sequence-specific, we conducted competition experiments using Fam-labelled centromeric DNA and unlabelled competitor DNA. We show that wild-type DNA is capable of competing out the labelled probe, while the CCG mutant cannot (Fig 7D), confirming the specific nature of the interaction. While Cep3 is present as a dimer in the core complex, there is only one CCG triplet conserved in the centromere. Crosslinking experiments have shown that there are other Cep3-DNA crosslinks

at a TGT triplet −12 to the CCG (Espelin *et al*, 1997), however a mutant at this site is unable to compete out labelled probe as also shown for the isolated Cep3 protein (Purvis & Singleton, 2008). Furthermore, a double mutant where both the CCG and TGT triplets are mutated has the same effect as the single CCG mutant. It therefore seems that Cep3 possesses only one DNA-binding site.

## Discussion

The data presented here provide new insights into the previously elusive Ctf13 protein, and the assembly of the CBF3 core complex. The requirement for using the native organism for recombinant expression of the complex supports the notion that a complex assembly pathway requiring post-translational modifications is required to form active CBF3. We find that expression of the complex is considerably improved by simultaneous over-expression of Sgt1, an adaptor protein involved in Hsp90-assisted protein maturation, supporting the previously described role of this chaperone in the CBF3 pathway (Bansal *et al*, 2004; Lingelbach & Kaplan, 2004). The recently described Sgt1-Skp1 structure (Willhoft *et al*, 2017) shows that Skp1 binds to Sgt1 and Cep3 using overlapping interfaces. Therefore, Sgt1 must dissociate from Skp1 before assembly of the core complex.

Although we were not able to build a full atomic model of Ctf13, the structural data we have are sufficient to unambiguously identify it as an F-box-leucine-rich-repeat protein, in the same family as the Skp2 component of the SCF[Skp2] E3 ubiquitin ligase. However, Ctf13 has considerably diverged from Skp2 with the addition of an expanded N-terminal domain around the F-box and an insertion in the repeats that comprises a second helical domain. Interestingly, the

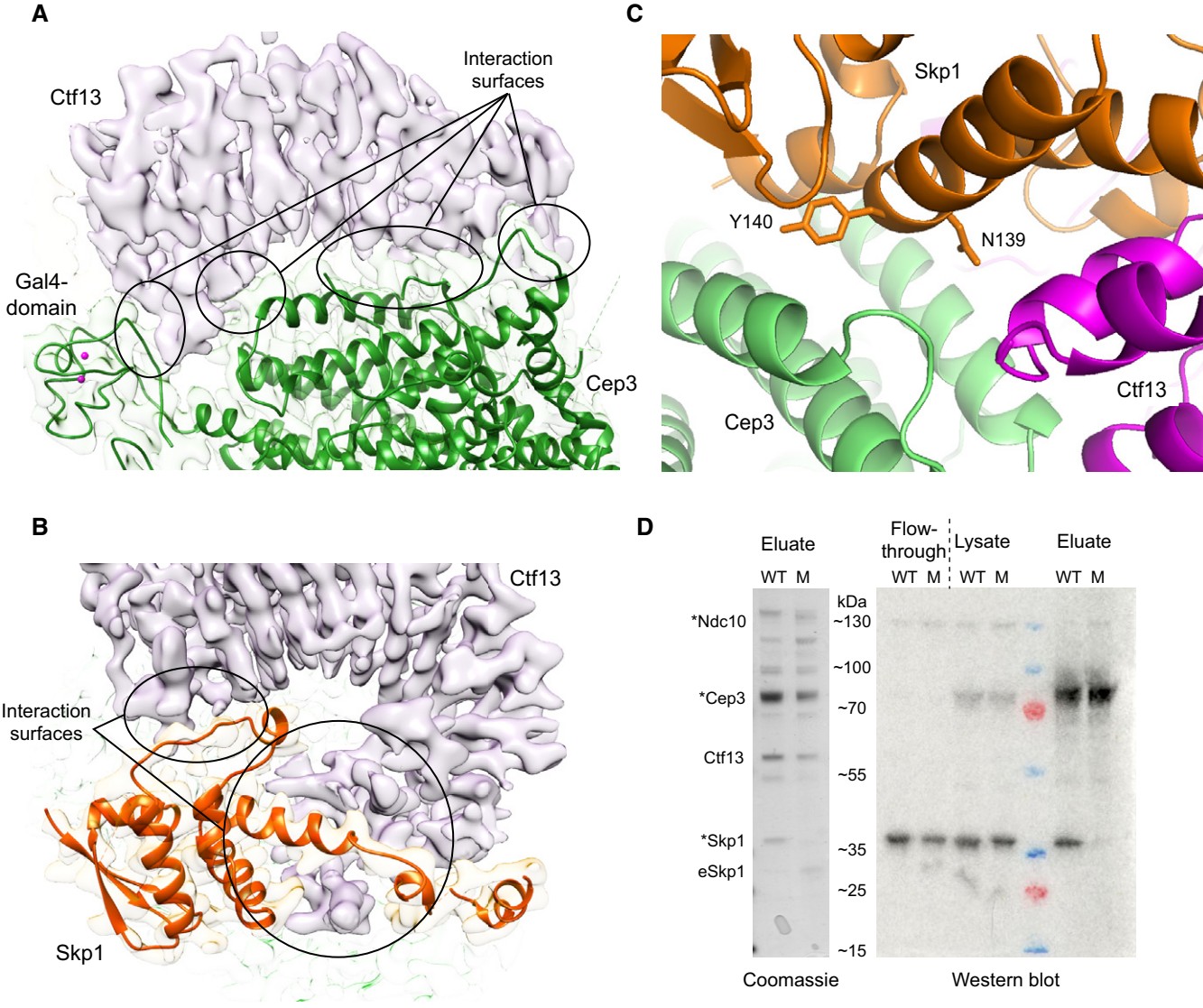

**Figure 6. Interaction of subunits Cep3, Ctf13 and Skp1.**

A  Extensive interaction surfaces (black circles) of Ctf13 and Cep3. Ctf13 is shown as density in pink and Cep3 as ribbon diagram and density in green.
B  Binding areas of Ctf13 to Skp1 (black circles). As above Ctf13 is shown as density in pink, and Skp1 as ribbon diagram and density in orange.
C  Detailed view of the interaction site of Cep3 and Skp1, highlighting involved residues (N139, Y140).
D  Interaction studies of Skp1 (N139K, Y140K) mutant. SDS–PAGE and Coomassie stain of eluates (left panel) show that only wild-type HA-tagged (WT), but not mutant Skp1 (M) can form a complex with the other subunits. Mutant HA-tagged Skp1 gets replaced by endogenous Skp1 (eSkp1), which migrates faster due to the lack of HA-tag (asterisks indicate HA-tagged subunits). The right panel shows the corresponding Western blot with anti-HA antibody, where no signal of mutant Skp1 could be detected in the eluates.

Ctf13 F-box/Skp1 interaction present in our structure is significantly different from all currently known examples, with substantially altered positioning of the helices in both the F-box and Skp1. It is notable that current F-box crystal structures have been determined using bacterially expressed and therefore non-phosphorylated proteins, while the phosphorylation status of Skp1 has been shown to affect binding to the F-box (Beltrao *et al*, 2012). Whether the differences observed in our structure relate to the phosphorylation status of Skp1 or are simply intrinsic to the CBF3 complex remain to be determined. Our structure also highlights the function of a previously unresolved loop between residues 105 and 112 in Skp1 in forming inter-subunit contacts. While this loop is widely conserved, it is not clear whether its role in protein–protein interactions is specific to CBF3 or might be a more general feature of Skp1-containing protein complexes. Our structure also highlights the remarkable repurposing of protein folds from unrelated pathways by CBF3: ubiquitination in the case of Skp1 and Ctf13 (SCF ligase), transcription by Cep3 (Gal4), and DNA modification by Ndc10 (Topoisomerase IB). Given the ancient origins and widespread occurrence of these cellular processes, it is tempting to view the occurrence of point centromeres

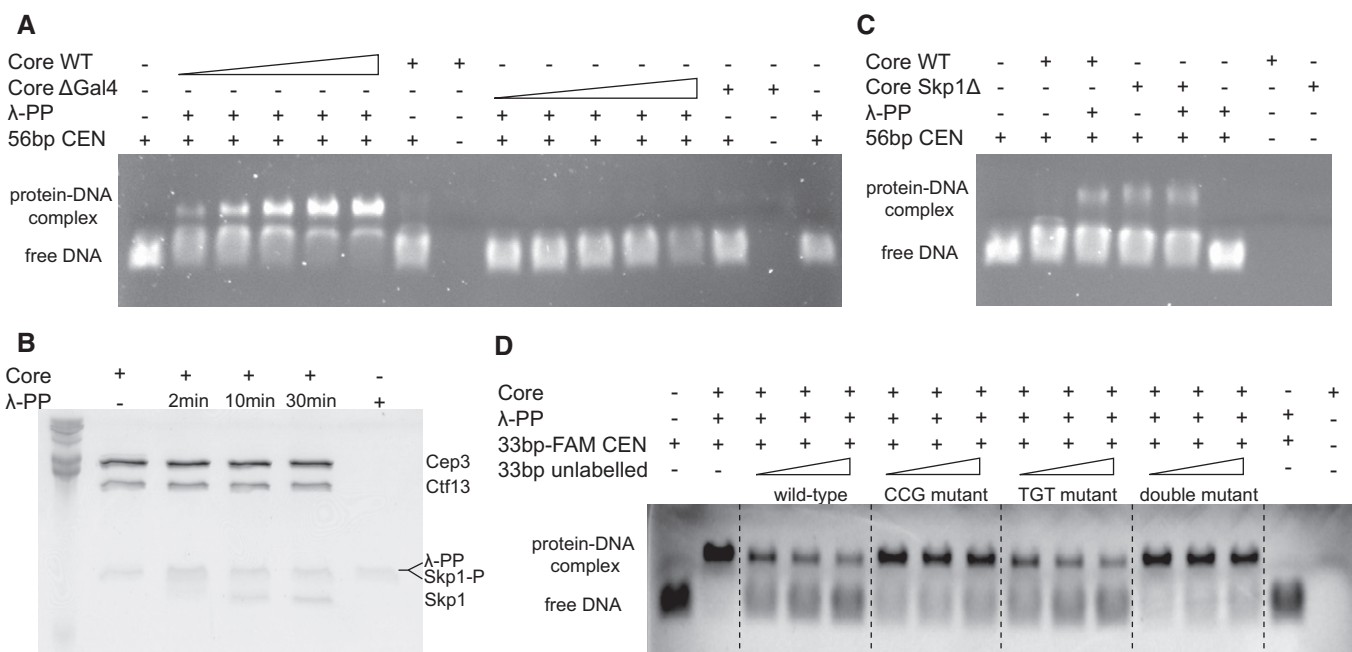

**Figure 7.  DNA-binding studies of CBF3 core complex and mutants.**

A   Electrophoretic mobility shift assay (EMSA) comparing the DNA-binding activities of wild-type core complex (coreWT) and core complex lacking the N-terminal Gal4-domain of Cep3 (coreΔGal4). Increasing concentrations (0.3–1 μM) of either coreWT or coreΔGal4 were mixed with 0.1 μM 56 bp CEN3 DNA (Appendix Table S3). 200 units of λ-protein phosphatase (λ-PP) were added, and the reaction was incubated at 30°C for 30 min. Free DNA (lane 1), reactions without λ-phosphatase (lanes 7 and 14), protein alone (lanes 8 and 15) and DNA with λ-phosphatase (lane 16) were run as controls.

B   Phos-tag–SDS–PAGE/Coomassie stain to demonstrate which subunit of CBF3 core complex is dephosphorylated by λ-phosphatase. A 12% SDS–polyacrylamide gel with 100 μM Phos-tag AAL-107 was prepared and core complex without λ-phosphatase (lane 2), core complex pre-treated with λ-phosphatase for 2, 10 and 30 min (lanes 3–5) and λ-phosphatase alone (lane 6) were run at 220 V for 45 min. Phos-tag ligand has a phosphate affinity resulting in a mobility shift of phosphorylated proteins.

C   EMSA showing the λ-phosphatase dependency of the core complex to bind to 56 bp CEN3 DNA, compared to a mutant construct (core Skp1Δ), which has a deletion of a N-terminal loop of Skp1 (residues 37–64). 1 μM protein was mixed with 0.1 μM DNA and, if applicable, 200 units of λ-phosphatase were added. Free DNA (lane 1), protein alone (lanes 7 and 8) and DNA with λ-phosphatase (lane 6) were run as controls.

D   EMSA to visualise the sequence specificity of DNA binding of the core complex. 0.7 μM core complex was mixed with 0.1 μM FAM-labelled 33 bp CEN3 DNA and increasing concentrations (0.4, 0.8, 1.2 μM) of unlabelled competition DNA. As competition DNA either wild-type CEN3 DNA (wild-type), CEN3 DNA with the CCG triplet mutated (CCG mutant), CEN3 DNA with the TGT-triplet mutated (TGT mutant), or both CCG and TGT-triplet mutated (double mutant) was used.

and CBF3 as a relatively recent innovation, which fits with some proposals about centromere evolution and point centromeres representing a modification of a plasmid partitioning system, arising only in certain yeasts (Malik & Henikoff, 2009).

We show that Skp1 and the Ctf13 F-box directly contact Cep3, causing stabilisation and inactivation of the Gal4-domain of one of the Cep3 monomers, while the other remains free. We further show that the core complex is capable of binding to DNA in a sequence-specific manner and that the free Gal4-domain of Cep3 is likely responsible. Intriguingly, we find that there could also be some form of regulation of DNA binding, which depends on dephosphorylation of the Skp1 subunit via an unidentified phosphatase. There is an extended, flexible loop between residues 37 and 64 present in the budding yeast Skp1 protein which contains several phosphosites. Deletion of this loop (Skp1Δ) allows constitutive DNA binding by the core complex, implicating it in this regulatory process. Interestingly, *in vivo* studies of Skp1Δ show it can rescue the Skp1 null mutant (Connelly & Hieter, 1996) and deletion of phosphosites results in viable cells (Stemmann *et al*, 2002). These results could be explained if indeed the only result of the loop phosphorylation is to inhibit DNA binding. With the current structural data, it is

difficult to assess how the phosphorylation status of this loop could influence DNA binding. We suggest two possible mechanisms: DNA bound by the free Gal4-domain lies across the core complex in a way that brings it into close proximity of the 37–64 loop and the negative charge of the phosphorylations inhibits DNA binding. Alternatively, dephosphorylation of the loop might lead to a conformational change in the complex which allows DNA to bind stably. It is striking that the central channel of the complex could accommodate duplex DNA nicely by a relatively small movement of the Skp1/Ctf13 heterodimer. It should be noted that negative stain EM analysis of the complex containing loop-deleted Skp1 (which effectively mimics the dephosphorylated state in our assays) appears indistinguishable from the wild type, suggesting major conformational changes are unlikely. It is also notable that phosphorylation, not dephosphorylation, has been proposed as being necessary for assembly of active CBF3 (Lechner & Carbon, 1991; Kaplan *et al*, 1997). These findings might be reconciled if differential phosphorylation of Ctf13 and Skp1 is required to assemble and activate the complex, possibly in a cell cycle-dependent manner. We have summarised our mechanistic findings and possible interactions within the full complex in a simple model (Fig 8).

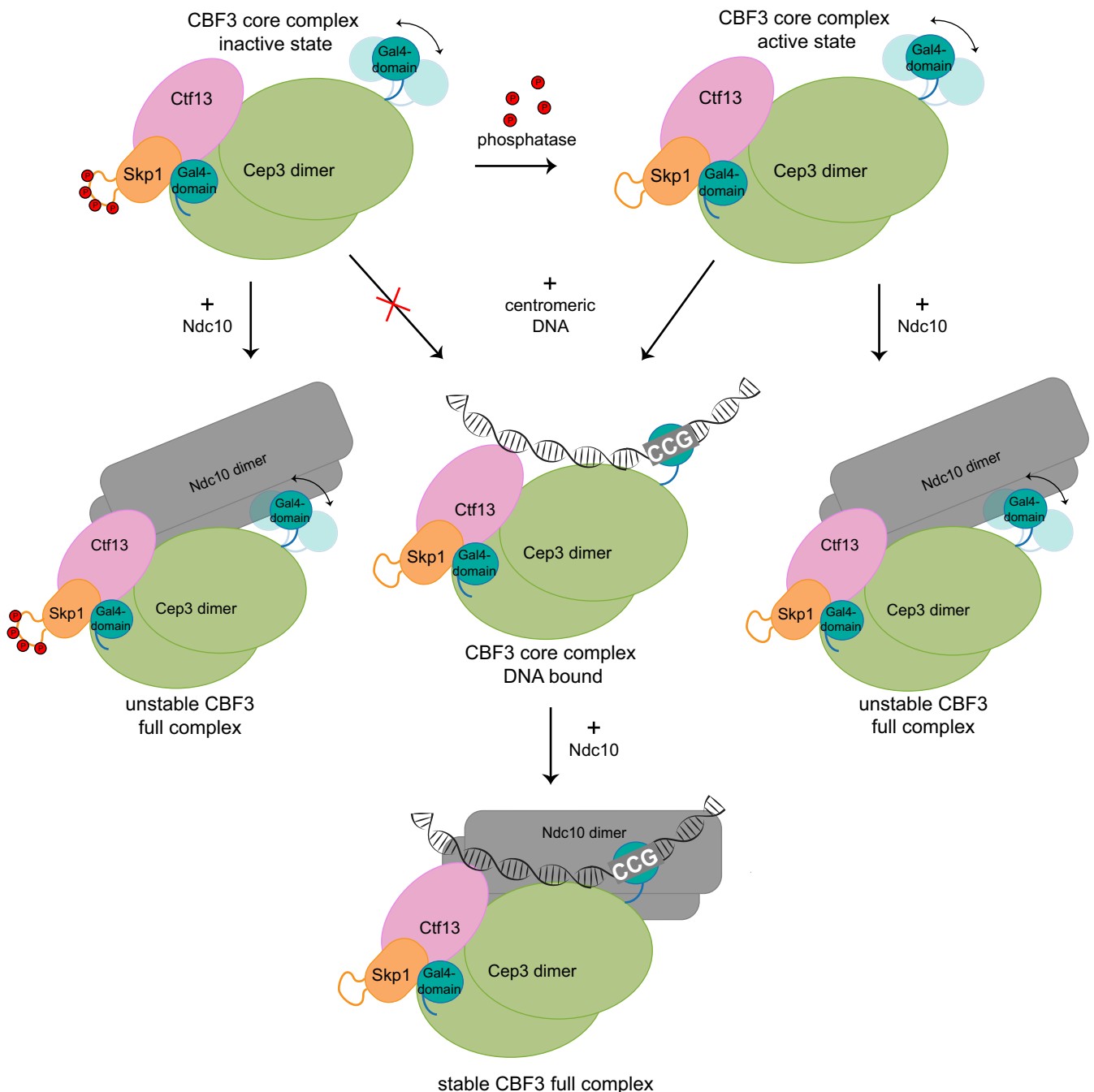

**Figure 8. Schematic model of CBF3 assembly and DNA binding.**

The formation of the CBF3 core complex, consisting of a homodimer of Cep3 and a heterodimer of Skp1/Ctf13, leads to the sequestering and inactivation of one of the two DNA-binding Gal4-domains, while the other remains free (CBF3 core complex—inactive state). Dephosphorylation of the Skp1 37–64 loop activates the DNA-binding activity of the complex. The free Gal4-domain binds to the CCG triplet of the centromere (CBF3 core complex—DNA bound). An Ndc10 dimer can then bind the core complex and associate with DNA in a sequence-independent manner.

It also remains to be established how Ndc10 binds to the core complex. As previously shown, the N-terminus of Ndc10 (NTD, residues 1–551) is needed for binding to the core (Cho & Harrison, 2011). However, the NTD alone seems to be less stably bound than the full-length protein. Negative stain EM analysis of the full complex has shown that although full-length Ndc10 binds

the core complex stably enough to withstand size-exclusion chromatography, Ndc10 is mostly unstructured. This suggests that additional factors are needed to fully fold Ndc10. One likely factor is DNA. Although further studies are needed to prove this assumption, it is notable that expression of CBF3 Skp1Δ, which comprises the active DNA-binding state, can pull out more Ndc10

than the wild-type complex. Purified CBF3 Skp1Δ, with no residual DNA present, behaves identically to wild-type CBF3, suggesting that it is not the difference of Skp1 which leads to this effect. The fact that Ctf13 appears somewhat disordered in our structure suggests an additional factor is required to stabilise it, which may well be Ndc10.

It is an open question as to how a large complex such as CBF3 and a centromeric nucleosome can simultaneously occupy a relatively short centromere sequence. Our study suggests that the core complex binds to the centromere via the "free" Gal4 domain in one Cep3 monomer. This is linked to the rest of Cep3 by a relatively long linker, which would allow the core considerable freedom of movement relative to the DNA. If the major contact between the Cse4 nucleosome and CBF3 is via Ndc10 as previously proposed (Cho & Harrison, 2011), the core complex could sit just outside the nucleosome, and position Ndc10 at the DNA entrance/exit directly adjacent to the histones. Such an arrangement would allow considerable flexibility facilitating nucleosome assembly steps and conceivably unusual centromeric DNA topologies (Henikoff *et al*, 2014). Further structural and biochemical studies will be required to address these questions.

# Materials and Methods

### Molecular biology

Synthetic genes codon-optimised for expression in *S. cerevisiae* encoding the four CBF3 subunits were purchased from GeneArt and cloned into inducible expression vectors kindly provided by John Diffley. A double-StrepTactin tag was added to the C-terminus of Cep3 for purification purposes.

### Protein expression and purification

The four CBF3 genes, as well as those coding for Sgt1 and Gal4, were transformed into *S. cerevisiae* using standard techniques. Cells were grown in YP-raffinose media and induced with 2% galactose at 30°C, shaking at 200 rpm for 4 h. Cells were harvested by centrifugation and re-suspended in Buffer A (50 mM HEPES, pH 7.5, 250 mM NaCl, 10% glycerol, 0.02% NP-40, 1 mM DTT) complemented with protease and phosphatase inhibitors. Cells were frozen drop-by-drop and lysed at LN2 temperatures with a SPEX freezer mill® 6875D (AXT). Cell debris was removed by centrifugation at 30,000 *g*, 4°C, for 50 min. The protein complex was purified by affinity chromatography (Strep-Trap HP, GE Healthcare), followed by HiTrap Heparin HP (GE Healthcare) to separate CBF3 core complex from Ndc10. CBF3 core and Ndc10 were either kept separately or combined depending on downstream procedures. TEV protease was added and the double-StrepTactin tag cleaved at 4°C overnight. Proteins were further purified with anion exchange chromatography (Poros Q, Thermo Fisher), followed by size-exclusion chromatography (Superose 6, 10/100, GE Healthcare) in the final buffer of 10 mM HEPES, 300 mM NaCl and 1 mM DTT. Protein purity was assessed by SDS–PAGE and Coomassie stain, and the identity of the individual bands was confirmed by mass spectrometry. All described mutants were expressed and purified with the same procedure.

### Negative stain EM

Appropriate dilutions of protein were pipetted onto glow-discharged continuous carbon grids and stained with uranyl acetate following standard procedures. Particles were visualised with a 120 kV Tecnai G2 microscope and Gatan Ultrascan 2 camera.

### Sample preparation for cryo-EM

Purified CBF3 core complex was diluted to a final concentration of 0.1 mg/ml immediately after final size-exclusion chromatography. 3 μl of sample was pipetted onto a glow-discharged cryo-EM grid (Quantifoil Cu 2/2, 300 mesh) and incubated for 30 s. Excess protein was blotted away with filter paper and flash-frozen in liquid ethane using a manual plunge freezer. Grids were stored in LN2 until data collection.

### Data collection and image processing

Data were collected on a Titan Krios electron microscope (FEI) at 300 kV equipped with a K2 Summit direct electron detector (Gatan) operated in counting mode. The imaging mode employed was nanoprobe EFTEM (energy-filtered TEM) and the energy filter operated in zero-loss peak mode with a slit width of 20 eV. Defocus range was set from −1.5 to −3.5 μm, and 25-frame movies were recorded with a total dose of 54 e/A$^2$. Frames were aligned with patch-correction, and dose-weighted summed images generated using MotionCor2 (Zheng *et al*, 2017). Contrast transfer function correction was carried out using GCTF (Zhang, 2016). Further image processing, including particle picking, was performed using Relion 2 (Fernandez-Leiro & Scheres, 2017).

Auto-picking of the 8,755 micrographs resulted in an initial set of ~3.5 million particles. Bad particles were removed using particle sorting, two rounds of 2D classification, and a single round of 3D classification resulting in a final data set of 209,751 particles. An initial model for the complex was generated from the PDB entry for Cep3Δ (PDB ID: 2veq), low-pass filtered to 40 Å resolution. The refined model was sharpened by applying a B-factor of −150 Å$^2$, and overall resolution was calculated using the Fourier shell correlation 0.143 criterion (Rosenthal & Henderson, 2003). Analysis of map anisotropy was carried out using 3DFSC (Tan *et al*, 2017). Crystal structures were manually placed into the cryo-EM density map and fitted as rigid bodies using Chimera (Pettersen *et al*, 2004). PDB codes are as followed: Cep3Δ: 2veq, Skp1: 1nex, Skp1-Skp2: 1fqv, Skp1-Cul1: 1ldk, Hap1: 2hap. Re-building of the model was carried out using Coot (Emsley & Cowtan, 2004) and real-space refinement of the models in phenix.refine (Adams *et al*, 2004). Local resolution was estimated using ResMap (Kucukelbir *et al*, 2013).

### DNA-binding assays

DNA fragments (Appendix Table S3) were purchased (Sigma Aldrich), and single strands were annealed by heating to 95°C and subsequent stepwise cooling. Purified protein complexes at given concentrations were mixed with CEN3 DNA fragments of specified length and concentration. 200 units of lambda protein phosphatase (NEB) and 1 mM MnCl$_2$ were added, and the reactions were incubated at 30°C, 30 min before electrophoresis on a 0.8% agarose gel

with Tris-borate-EDTA buffer. Agarose gels were chosen as the protein complex was not able to penetrate an acrylamide gel, possibly due to its size.

### *In vitro* interaction studies

All four CBF3 subunits (either wild-type or mutants, as specified) were expressed as described above, with an additional HA-tag on all genes but Ctf13. An equal amount of cells was lysed, cell debris removed by centrifugation and proteins pulled out with a Strep-Trap HP (GE Healthcare) column. Lysate and elution were run on a SDS–PAGE and transferred to a nitrocellulose membrane using the iBlot™ gel transfer device (Invitrogen). Blots were probed with anti-HA antibody, which was detected with a horseradish peroxidase-conjugated secondary antibody (GE Healthcare).

### Mobility shift assay with Phos-tag

12% polyacrylamide SDS gels were prepared according to manufactures recommendations. 100 μM Phos-tag™ and 200 μM $MnCl_2$ were added to the separating gel to allow for a band shift of phosphorylated proteins. Dephoshorylation reactions were set up as described above (DNA-binding assays), and reactions were stopped at specified time points by adding SDS-loading buffer and boiling the sample. 10 μl of the samples was loaded and run at 220 V for 45 min. Gels were stained with InstantBlue™ (Expedeon).

### Mass spectroscopy

Protein molecular mass was determined using a microTOFQ electrospray mass spectrometer (Bruker Daltonics, Coventry, UK). Protein was desalted using a 2 mm × 10 mm guard column (Upchurch Scientific, Oak Harbor WA) packed with Poros R2 resin (PerSeptive Biosystems, Framingham). Protein was injected onto the column in 10% acetonitrile, 0.10% acetic acid, and washed with the same solvent and eluted in 60% acetonitrile, 0.1% acetic acid. Desalted protein was then infused into the mass spectrometer at 3 μl/min using an electrospray voltage of 4.5 kV. Mass spectra were deconvoluted using maximum entropy software (Bruker Daltonics, Coventry, UK).

### Multi-angle light scattering

200 μg of core complex was injected onto a Superose 6 Increase 10/300 column (GE Healthcare) coupled to a Wyatt Dawn 8+ MALS system. Data were analysed using Astra software.

### Data availability

The final EM density map has been deposited in the EMDB database under accession code EMD-4163 and the refined Cep3 and Skp1 structures in the PDB under accession code 6F07.

**Expanded View** for this article is available online.

### Acknowledgements

We wish to thank the Crick Electron Microscopy Platform for access to microscopes, R. Carzaniga and T. Pape for assistance with EM data collection, the Crick Structural Biology Platform for computing support and Krios access, J. Diffley for supplying plasmids, C. Bouchoux for advice on yeast expression, S. Mouilleron for assistance with SEC-MALS, C. Davis for intact mass analysis and A. Costa for extensive discussions on cryo-EM methodology. Initial Cryo-EM data were collected at the UK national electron bio-imaging centre (eBIC) (proposal EM-14294), funded by the Wellcome Trust, MRC and BBSRC. This work was supported by the Francis Crick Institute, which receives its core funding from Cancer Research UK (FC001155), the Medical Research Council (FC001155) and the Wellcome Trust (FC001155).

### Author contributions

The study was designed by VL and MRS. EM data collection was carried out by VL and AN. Image processing was carried out by VL and MRS. All other procedures were carried out by VL. The manuscript was written by VL and MRS.

### Conflict of interest

The authors declare that they have no conflict of interest.

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
