## [Review Process File · The EMBO Journal]

Structural Basis for Assembly of the CBF3 Kinetochores Complex

Vera Leber, Andrea Nans, and Martin R Singleton

Review timeline:

Submission date:	31 August 2017
Editorial Decision:	5 October 2017
Revision received:	30 October 2017
Accepted:	20 November 2017

Editor: Hartmut Vodermaier

Transaction Report:

1st Editorial Decision

5 October 2017

Thank you for submitting your manuscript on the CBF3 complex cryo-EM structure to The EMBO Journal. We have now received the attached reports of three expert referees, who all consider this structure an important achievement and significant advance in principle. We shall therefore be happy to publish the study in The EMBO Journal pending adequate addressing of various specific points raised in the reports. As you will see, most of these points relate to alternative presentation, deeper mining of the available data, and further discussion and interpretation in light of the current standing of the field. In addition, there are also some technical questions and queries regarding the apparent phospho-regulation, which I feel would be important to address. On the other hand, while further EM or cross-linking mass-spec results on the full complex (as suggested by referee 2) would certainly be an additional asset, I realize that these might require extensive extra time and effort and may therefore not fall within the scope of this revision.

I would therefore like to invite you to prepare a revised manuscript and response letter to the referee comments, keeping in mind that our policy to allow only a single round of major revision will make it important to carefully answer to all points raised at this stage. Should you have any additional questions/comments regarding the referee reports or the revision requirements, please therefore do not hesitate to get in touch with me ahead of resubmission. If needed, we might also extend the revision period, during which publication of any competing work elsewhere would have no negative impact on our final assessment of your own study.

Please refer to the sections below for additional information on preparing, formatting and uploading a revised manuscript, adherence to which should greatly facilitate the processing of the resubmitted manuscript towards publication.

Thank you again for the opportunity to consider this work for The EMBO Journal. I look forward to your revision.

REFeree REPORTS

Referee #1:

The CBF3 complex of *S. cerevisiae* is required for DNA-sequence specific recognition of the centromere. It is not conserved in most other eukaryotes, but it plays a crucial function for kinetochore assembly and stability in budding yeast, a widely used model system for kinetochore studies. CBF3 has been largely impervious to reconstitution and structural biology investigations. Here, Leber and Singleton report a very important step forward in the study of CBF3, the development of a good recombinant strategy for its expression. This allowed them to isolate a stable "core" subcomplex of CBF3 including Skp1, Ctf13, and Cep3, but devoid of the less tightly bound subunit Ndc10. This core complex was amenable to a high-resolution EM analysis that delivered important insights not only on the overall organization of the CBF3 core, but also on the structural organization of Ctf13, an F-box subunit which had escaped structural analysis so far (while parts of the structures of Skp1 and Cep3 had been known).

In general, I feel that the work is interesting and important, and I consider it a milestone, first of all on the basis of the significant profusion of efforts from several laboratories towards obtaining a structure of CBF3 and the relatively little reward that has come to it. Here persistence has been the key. In addition, I believe that the work described here may represent a milestone for future work of reconstitution and structural analysis of larger parts of the *S. cerevisiae* kinetochore.

The two main elements of novelty in the manuscript are the structure of Ctf13, which is shown to be a structurally divergent leucine-rich repeat F-box protein, and the overall organization of the core complex, which revealed the basis of its assembly and stoichiometry. I feel that this will be of interest to a significant family of researchers and I am therefore very supportive of publication of this manuscript.

I don't have any major issues with this manuscript. The only question regards Figure 7D, where the authors have not excluded the possibility that a combination of the two mutations in a single oligo (with both the CCG and TGT triplets mutated) may create a stronger effect than that observed with only Mut1. (This gel is very dark and I would recommend replacing it if possible.) An asymmetric Cep3 dimer with two distinct subunits, one carrying the Gal4 domain and the other not, would be another interesting way to address this question, but I realize that this may be beyond the scope of this manuscript.

Minor points

Page 3 '...such as short sequence...' Correct 'as'

Page 3 'principle' should read 'principal'

Page 11 '...previously mysterious...' Maybe use "elusive"

Page 12 '...causing to stabilization...' Remove 'to'

Page 12 '...that that...'

Page 13 '...remains to establish...' Maybe better '...remains to be established...'

Referee #2:

The manuscript "Structural Basis for Assembly of the CBF3 Kinetochore Complex" by Leber and Singleton describes a cryo-EM structure of the *S. cerevisiae* CBF3 "core complex" comprising two copies of Ctf13 and one copy each of Skp1 and Cep3. Together with the additional CBF3 complex subunit Ndc10, this complex is the main sequence-specific DNA binding factor in the budding-yeast kinetochore, and aids the recruitment and positioning of a single Cse4/CENP-A nucleosome to establish the kinetochore. The current manuscript demonstrates purification of both the full CBF3 complex, with Ndc10, and the "core complex" without Ndc10. A 3.6 Å cryo-EM structure of the core complex reveals its overall architecture, and reveals for the first time that Ctf13 is an F-box/leucine rich repeat (FBXL) protein, though the resolution is not high enough to build a full model of this protein. This work represents an important contribution to understanding the establishment of the budding-yeast kinetochore, and I am in principle supportive of publication in EMBO J. However, the analysis of the data and its implications is rather light at this point. I would ask that the authors perform a few relatively obvious follow-up experiments that would greatly extend this work, or alternatively address why these experiments are not feasible (points 1 and 2 below). I would also ask for more complete

analysis and description of the results in hand (point 3 below).

Major points:

1) The authors purified both the full CBF3 complex with Ndc10, and the "core" complex without Ndc10. Given that they clearly possess the capability to perform high-resolution EM, and the full complex looks to be of similar purity to the partial complex, why isn't structural analysis of the full complex presented or at least discussed? It seems that particle heterogeneity might be a problem, as Ndc10 is prone to dissociate from the complex and it contains large regions of predicted disorder. However, Cho & Harrison (2011 NSMB) showed that Ndc10 binds the CBF3 core through its ordered N-terminal domain, so perhaps this analysis could work even if large portions of the Ndc10 C-terminal region were disordered. If analysis of the full complex were possible, even at low resolution, this would greatly improve the impact of the manuscript.

1a) If EM analysis of the full CBF3 complex is impossible, have the authors considered cross-linking mass spec to identify pairs of residues in close proximity in the full complex? Given the structural knowledge now available for all components, this could provide insight into the overall architecture of the full CBF3 complex.

2) The authors note that DNA binding by the CBF3 core complex depends on pre-treatment with a phosphatase, suggesting that phosphorylation of the complex acts as a mechanism to control CBF3's DNA binding ability. This is interesting and worth pursuing in future work, perhaps by mapping the sites and measuring their cell cycle-dependent phosphorylation. For this work, the authors note that deletion of a specific loop in Skp1 rescues DNA binding, suggesting that the relevant phosphorylation sites are in this loop. Two questions on this: First, is there any evidence of phosphorylation in your electron density maps? The phosphorylation must be nearly quantitative for it to have such a strong effect on DNA binding, and therefore you may see the phosphates in your maps. Second, the authors note in the Methods that they verified the proteins by mass spec - is there evidence in these data of phosphorylation at specific sites? Bonus question (for future work): Is there a difference in conformation of the complex (as measured by EM) before and after phosphatase treatment?

3) EMBO J. allows 9 main figures. I would strongly urge the authors to put together a model figure that shows how their work has refined the models of how CBF3 binds centromeric DNA and nucleates the assembly of a kinetochore. This should also be accompanied by a more complete discussion of how the authors think Ndc10 integrates into this complex, how DNA is bound (and perhaps wrapped) by the different components of the complex, and crucially how CBF3 helps recruit and position a Cse4 nucleosome (this last point is mentioned by the authors in the introduction, but never addressed in the discussion). Such a figure and accompanying discussion would make the work accessible to a wider audience of readers.

4) How does the discovery that Ctf13 is an FBXL protein inform ideas of how the budding-yeast "point centromere" and accompanying specialized kinetochore structure evolved?

Minor points

Introduction, page 2. The authors should consider citing Westermann et al (Annual Review of Biochemistry 2007) along with the Santaguida & Musacchio 2009 review.

Page 3, line 2: "such as short" should be "such a short"

Results, Page 4: The authors note in the discussion that CBF3 complex expression was aided by simultaneous expression of Sgt1; this should be mentioned in the results.

Page 12, last paragraph, first line: "causing to stabilisation" should be "causing stabilisation"

Figure 1 could be improved by addition of size-exclusion chromatography data, and SEC-MALS light-scattering data if available, for both complexes.

Figure 2: This figure would be easier to understand if it were accompanied by a domain diagram (colored consistently with the structures) of each component, with known/fit domains labeled and

missing domains noted. Referencing later data, this diagram could also show the Skp1 loop the authors show in Figure 4, and the loop deleted in the DNA binding analysis. Finally, as different views are shown in each panel of Figure 2, a visual indication of the relevant rotations between them (as in Figure 3A) would be helpful.

Figure 3: As in figure 2, a detailed domain diagram of Ctf13 (perhaps with notes as to which domains contact other subunits in the complex) would aid understanding of this figure, and help a reader determine how much of the Ctf13 protein the authors have managed to build.

Referee #3:

In their study "Structural Basis for Assembly of the CBF3 Kinetochores Complex", Leber and Singleton utilize single particle cryo-electron microscopy (cryo-EM) structural analysis and complimentary biochemical assays to analyze the structure and function of the yeast CBF3 complex. CBF3 is required to recognize the sequence of point centromeres and deposit Cse4 nucleosomes, thereby templating kinetochores assembly. The structure and mechanisms of inner kinetochores proteins remain relatively mysterious, as well as the level of conservation at the level of molecular mechanism despite divergence of kinetochores architecture and specific protein components. This study makes an important contribution, providing significant new mechanistic insight into assembly and regulation of the yeast inner kinetochores.

The authors successfully reconstitute and purify the CBF3 complex, which is composed of a stably-associating "core" of Cep3, Ctf13, and Skp1. Their cryo-EM structure demonstrates that the Ctf13 regulatory subunit, which has not previously been structurally characterized, adopts an FBXL fold which forms a bridging interface between the Cep3 dimer and Skp1. Furthermore, the authors provide a compelling structural rationale for why the complex forms a 1:1:2 Ctf13:Skp1:Cep3 stoichiometry, which is linked to why Cep3 can only engage a single piece of centromeric DNA despite being a dimer. Ctf13 and Skp1 embrace one of the 2 N-terminal Gal4 domains of Cep3, locking it in a stable orientation and sterically occluding its DNA-binding site. Only one such interface can be formed, as a second Ctf13/Skp1 pair would be too large to engage the second Gal4 domain, which remains unobstructed and flexible, and is thus available to bind DNA. Thus, steric occlusion plays a key role in both symmetry breaking during complex assembly and regulation of DNA binding. Finally, the authors find that Skp1 phosphorylation by an unknown kinase regulates DNA engagement by an unresolved mechanism. In sum, this study provides substantial mechanistic insight, and points the way towards an exciting future direction for understanding CBF3 regulation by dissecting the structural and functional effects of Skp1 phosphorylation.

The writing of the manuscript is very clear and lucid. Furthermore, the authors are to be commended for their transparent presentation and interpretation of their structural data. This includes consistent presentation of density maps in their figures and their general interpretation at the appropriate level of detail with atomistic models given the varying resolution, particularly building poorly ordered sections of Ctf13 as poly-alanine. I believe this represents a very nice piece of work, and should be accepted pending minor revisions, detailed below.

Significant points:

Map anisotropy and flexibility of Ctf13

1) The authors note, and the Euler angle density plot presented in Figure S1 supports, 2 preferred orientations of their particles in ice. While the FSC curve suggests an overall resolution of 3.6 Angstroms, it is likely that their resolution is anisotropic, which is apparent in some level of streaking in their density maps, particularly of Ctf13. It is now possible, and would be useful, to quantify this effect utilizing a tool recently released by NRAMM and the Lyumkis lab:

<https://github.com/nysbc/Anisotropy>

If indeed significant anisotropy is present, the authors could try "balancing" the views by excluding a fraction of the particles corresponding to preferred views. This could be implemented in practice by not including particles corresponding to a subset of 2D classes which correspond to these views using the subset selection tool in Relion. As the authors have a very large number of particles, it is possible that

this would improve the isotropy of the map without impacting the global resolution.

If this strategy fails to improve the map, it would be useful at a minimum to include the anisotropic resolution assessment in Figure S1.

2) It would also be worthwhile to attempt to improve the Ctf13 density through focused classification / signal subtraction, as described by Scheres and colleagues:

PMID: 26623517

Since the authors have nice atomic models for Cep3 and Skp1, it should be straightforward to segment their reconstruction using these models (using the "Color zone" / "split map" options in UCSF chimera would be a simple way) to generate a mask defining the region occupied by these molecules. This could then be used in the Relion GUI to subtract the signal corresponding to these subunits from aligned particles, which could then be subjected to 3D classification within an inverse mask corresponding to the region occupied by Ctf13, as described in detail here:

<ftp://ftp.mrc-lmb.cam.ac.uk/pub/scheres/relionreview2016.pdf>

It is possible this procedure will identify discrete conformational states adopted by Ctf13, or one better-ordered structural state, which could then be used to improve the atomic model.

For both points 1 and 2, I would explicitly note that failure to achieve improved maps through these procedures would not impact my overall positive assessment of this study. However, I do think it is important for the authors to make a bona fide attempt to achieve the best possible maps from their data using currently available tools.

Mechanism of CBF3 regulation via Skp1 phosphorylation

The inhibition of CBF3 by Skp1 phosphorylation is an important observation, and I was surprised that the authors did not make an attempt to rationalize it in the context of their structure, particularly since they honed in on a likely site of phosphorylation, the 37-64 loop.

1) It would be very useful to include a view of the likely phosphorylation loop (a panel in Figure 7 would be appropriate). My impression is that this loop likely faces away from the rest of the structure and thus its effects may be somewhat difficult to rationalize, but this is important information to guide future studies. The implications of the structure for understanding this phosphoregulation should be included in the discussion (at a minimum, what the structure rules out).

2) What is the phosphorylation status of the complex the authors utilized for their structural analysis? I'm guessing it was not lambda phosphatase treated? If so, it should be explicitly mentioned that the state visualized likely represents an inhibited state.

3) Since they have the material in hand and here do report the preliminary purification of a full CBF3 complex including Ndc10, I believe it would be appropriate to biochemically test and report if lambda phosphatase treatment strengthens the association between the core complex which was visualized and Ndc10 in this paper. If successful, this complex could be the subject of a follow-up study, which is beyond the scope of this manuscript.

Minor points

1) In Figure 4 C, it is difficult to keep track of which labelled alpha helix corresponds to which protein. Perhaps adding the first letter of that protein (e.g. S or C) would make this clearer. It also appears a label for Skp1 alpha helix 3 has been omitted.

2) In Figure 5 and in the text, the authors claim individual Zinc ions are resolved in their maps, which would be quite surprising at 3.6 Angstroms and is not supported by the density they present in 5b, where separated density peaks for each Zn ion are not apparent. This could be changed to "The C6Zn2 cluster is clearly visualized" in the text and "density for the cluster" in the legend without impacting the interpretation.

3) In figure 7D and the legend, the labelling for the mutants is a bit confusing. It is apparent that Mut1 is the actual mutant which produces a lack of competitive binding, while Mut2 is presumably a control which fails to compete. It would be better to label them in a manner which makes this explicit.

4) In Figure S1b, the various FSC curves which are presented are not indicated in the legend. Presumably these correspond to masked, unmasked, and phase-randomized plots generated by Relion. What is what should be explicitly indicated.

5) The Zheng et al. paper reporting MotionCor2 has been published in Nature Methods: PMID 28250466 . This should be cited rather than the BioRxiv preprint.

1st Revision - authors' response

30 October 2017

Referee #1:

*The CBF3 complex of *S. cerevisiae* is required for DNA-sequence specific recognition of the centromere. It is not conserved in most other eukaryotes, but it plays a crucial function for kinetochore assembly and stability in budding yeast, a widely used model system for kinetochore studies. CBF3 has been largely impervious to reconstitution and structural biology investigations. Here, Leber and Singleton report a very important step forward in the study of CBF3, the development of a good recombinant strategy for its expression. This allowed them to isolate a stable "core" subcomplex of CBF3 including Skp1, Ctf13, and Cep3, but devoid of the less tightly bound subunit Ndc10. This core complex was amenable to a high-resolution EM analysis that delivered important insights not only on the overall organization of the CBF3 core, but also on the structural organization of Ctf13, an F-box subunit which had escaped structural analysis so far (while parts of the structures of Skp1 and Cep3 had been known).*

*In general, I feel that the work is interesting and important, and I consider it a milestone, first of all on the basis of the significant profusion of efforts from several laboratories towards obtaining a structure of CBF3 and the relatively little reward that has come to it. Here persistence has been the key. In addition, I believe that the work described here may represent a milestone for future work of reconstitution and structural analysis of larger parts of the *S. cerevisiae* kinetochore.*

The two main elements of novelty in the manuscript are the structure of Ctf13, which is shown to be a structurally divergent leucine-rich repeat F-box protein, and the overall organization of the core complex, which revealed the basis of its assembly and stoichiometry. I feel that this will be of interest to a significant family of researchers and I am therefore very supportive of publication of this manuscript.

I don't have any major issues with this manuscript. The only question regards Figure 7D, where the authors have not excluded the possibility that a combination of the two mutations in a single oligo (with both the CCG and TGT triplets mutated) may create a stronger effect than that observed with only Mut1. (This gel is very dark and I would recommend replacing it if possible.) An asymmetric Cep3 dimer with two distinct subunits, one carrying the Gal4 domain and the other not, would be another interesting way to address this question, but I realize that this may be beyond the scope of this manuscript.

We have re-done the DNA-binding experiment with an oligo containing both mutations as suggested (figure 7D). The conclusion is essentially the same as our original one; that mutating the second (TGT) triplet has little or no effect on binding. The revised gel should also be clearer.

We considered making an asymmetric Cep3 dimer as suggested, but there are significant technical issues. While in principle we could do this by making a linked dimer as a single polypeptide, the relative positions of the N- and C- termini of the proteins means an extremely long linker would be required (~70 Å). It is also probable that the linker would occlude the likely DNA binding site, and have deleterious effects on the expression and formation of the recombinant complex. Purifying a heterodimer from a mixed population of homo- and heterodimers is also difficult due to the small size of the Gal4 domain. While an interesting experiment, we feel we cannot readily carry it out in the time available.

Minor points

Page 3 '*...such as short sequence...*' Correct 'as'
Page 3 '*principle*' should read '*principal*'
Page 11 '*...previously mysterious...*' Maybe use "*elusive*"
Page 12 '*...causing to stabilization...*' Remove 'to'
Page 12 '*...that that...*'
Page 13 '*...remains to establish...*' Maybe better '*...remains to be established...*'

All these points have been addressed in the revision.

Referee #2:

*The manuscript "Structural Basis for Assembly of the CBF3 Kinetochore Complex" by Leber and Singleton describes a cryo-EM structure of the *S. cerevisiae* CBF3 "core complex" comprising two copies of Ctf13 and one copy each of Skp1 and Cep3. Together with the additional CBF3 complex subunit Ndc10, this complex is the main sequence-specific DNA binding factor in the budding-yeast kinetochore, and aids the recruitment and positioning of a single Cse4/CENP-A nucleosome to establish the kinetochore. The current manuscript demonstrates purification of both the full CBF3 complex, with Ndc10, and the "core complex" without Ndc10. A 3.6 Å cryo-EM structure of the core complex reveals its overall architecture, and reveals for the first time that Ctf13 is an F-box/leucine rich repeat (FBXL) protein, though the resolution is not high enough to build a full model of this protein. This work represents an important contribution to understanding the establishment of the budding yeast kinetochore, and I am in principle supportive of publication in EMBO J. However, the analysis of the data and its implications is rather light at this point. I would ask that the authors perform a few relatively obvious follow-up experiments that would greatly extend this work, or alternatively address why these experiments are not feasible (points 1 and 2 below). I would also ask for more complete analysis and description of the results in hand (point 3 below).*

Major points:

1) The authors purified both the full CBF3 complex with Ndc10, and the "core" complex without Ndc10. Given that they clearly possess the capability to perform high-resolution EM, and the full complex looks to be of similar purity to the partial complex, why isn't structural analysis of the full complex presented or at least discussed? It seems that particle heterogeneity might be a problem, as Ndc10 is prone to dissociate from the complex and it contains large regions of predicted disorder. However, Cho & Harrison (2011 NSMB) showed that Ndc10 binds the CBF3 core through its ordered N-terminal domain, so perhaps this analysis could work even if large portions of the Ndc10 C-terminal region were disordered. If analysis of the full complex were possible, even at low resolution, this would greatly improve the impact of the manuscript.

Although we can purify the full CBF3 complex including Ndc10, there are serious issues with the stability of the complex and heterogeneity of the particles that prohibit EM analysis at present. To illustrate this, we have revised figure EV1 to include negative stain images of the core (EV1A) and full (EV1B) complexes. It can be appreciated that the particles for the full complex are extremely disorganized, and appear to show a "halo" of what we assume is unstructured protein around the periphery of the particle, making any kind of classification or averaging impossible. In agreement with Cho & Harrison, we can also purify the core complex with the N-terminal domain of Ndc10 bound (EV1C). However, the binding is sub-stoichiometric, and the gel filtration profile poor (EV1D). The resulting particles also show substantial heterogeneity, with most appearing to be core only (EV1E). For these reasons, we have not been able to extend the analysis to the full complex, although this is clearly a desirable long-term aim.

1a) If EM analysis of the full CBF3 complex is impossible, have the authors considered cross-linking mass spec to identify pairs of residues in close proximity in the full complex? Given the structural knowledge now available for all components, this could provide insight into the overall architecture of the full CBF3 complex.

We have tried to carry out cross-linking/mass-spectrometry analysis of the complex as suggested. Although our control experiments demonstrated our protocols were working correctly, we have been unable to obtain reliable data for the full CBF3 complex. This probably results from the instability of Ndc10 binding as described above. We also note that the apparent extensive disorder of the Ndc10 C-terminal would likely give rise to many nonspecific cross-links making validation problematic. We should also point out that as we do not currently have a full, sequence-assigned model for Ctf13, any cross-links to this protein (which we think represents the main binding site for Ndc10) would not necessarily be locatable on the structure.

2) The authors note that DNA binding by the CBF3 core complex depends on pre-treatment with a phosphatase, suggesting that phosphorylation of the complex acts as a mechanism to control CBF3's DNA binding ability. This is interesting and worth pursuing in future work, perhaps by mapping the sites and measuring their cell cycle-dependent phosphorylation. For this work, the authors note that deletion of a specific loop in Skp1 rescues DNA binding, suggesting that the relevant phosphorylation sites are in this loop. Two questions on this: First, is there any evidence of phosphorylation in your electron density maps? The phosphorylation must be nearly quantitative for it to have such a strong effect on DNA binding, and therefore you may see the phosphates in your maps. Second, the authors note in the Methods that they verified the proteins by mass spec - is there evidence in these data of phosphorylation at specific sites? Bonus question (for future work): Is there a difference in conformation of the complex (as measured by EM) before and after phosphatase treatment?

We cannot see evidence of phosphorylation in the density maps. We should clarify that there are two loops in Skp1; from 37-64, which is the "DNA" loop, and a second, from 105-112. The 37-64 loop is not actually resolved in our EM maps (and is deleted from crystal structure constructs). We have modified the text and included a schematic of the sequences to make this clearer (figure 2E). The second loop, though not seen in crystal structures is resolved in our map, and helps mediate complex formation as described. In addition, at the resolution of the reconstruction it might be difficult to see phosphorylation directly, especially on mobile loops, where local resolution is often poor anyway. Intact mass analysis of the complex shows that there are indeed four phosphates on Skp1 (Appendix Figure S4), consistent with the results obtained previously (Stemmann et al., 2002). We attempted to confirm that they were located on the specific loop by analysis of tryptic digests, but this is rather complicated as the sequence bias of the loop results in very long peptides which are extremely difficult to directly detect in the multiply-phosphorylated state. Indirect evidence from the mass spec. strongly supports the phosphorylation of this specific loop (for example, no phosphorylation is seen on any other visible sites in Skp1) but given the complexity of the data and analyses we have not included them here. We feel that the intact mass results and experiments of Stemman et al. answer this question unambiguously. We have analysed the Skp1 Δ construct which effectively biochemically mimics the phosphatase-treated complex and found no obvious differences in structure (Appendix Figure S3). A high-resolution structure of this could be carried out in future.

3) EMBO J. allows 9 main figures. I would strongly urge the authors to put together a model figure that shows how their work has refined the models of how CBF3 binds centromeric DNA and nucleates the assembly of a kinetochore. This should also be accompanied by a more complete discussion of how the authors think Ndc10 integrates into this complex, how DNA is bound (and perhaps wrapped) by the different components of the complex, and crucially how CBF3 helps recruit and position a Cse4 nucleosome (this last point is mentioned by the authors in the introduction, but never addressed in the discussion). Such a figure and accompanying discussion would make the work accessible to a wider audience of readers.

We have included a schematic in figure 8 which summarises our findings and presents a speculative model of how the CBF3 complex forms and binds DNA. Given that there are several major unknowns about the process, particularly with regard to Ndc10, we have tried to avoid excessive and potentially misleading detail. We have expanded our views on how the complex may function and recruit the Cse4 nucleosome in the discussion.

4) How does the discovery that Ctf13 is an FBXL protein inform ideas of how the budding-yeast "point centromere" and accompanying specialized kinetochore structure evolved?

The finding that Ctf13 is also essentially a co-opted ubiquitin ligase component is interesting, and consistent with similar re-use of pre-existent protein folds to form CBF3. We have expanded on this point a bit in the discussion.

Minor points

Introduction, page 2. The authors should consider citing Westermann et al (Annual Review of Biochemistry 2007) along with the Santaguida & Musacchio 2009 review.

We have included this reference.

Page 3, line 2: "such as short" should be "such a short"

Results, Page 4: The authors note in the discussion that CBF3 complex expression was aided by simultaneous expression of Sgt1; this should be mentioned in the results.

Page 12, last paragraph, first line: "causing to stabilisation" should be "causing stabilisation"

Above mentioned points have been addressed.

Figure 1 could be improved by addition of size-exclusion chromatography data, and SEC-MALS light-scattering data if available, for both complexes.

Figure 1 has been revised by including a size-exclusion chromatography trace (Figure 1B) of the full complex and the core complex. We have carried out a SEC-MALS analysis of both complexes. The value obtained for the core was consistent with the predicted masses (Figure EV3C). We were unable to obtain clear data for the full complex, again probably relating to the issues described in response to point 1.

Figure 2: This figure would be easier to understand if it were accompanied by a domain diagram (colored consistently with the structures) of each component, with known/fit domains labeled and missing domains noted. Referencing later data, this diagram could also show the Skp1 loop the authors show in Figure 4, and the loop deleted in the DNA binding analysis. Finally, as different views are shown in each panel of Figure 2, a visual indication of the relevant rotations between them (as in Figure 3A) would be helpful.

We have included a schematic diagram of the four CBF3 subunits, and updated the main figure as requested.

Figure 3: As in figure 2, a detailed domain diagram of Ctf13 (perhaps with notes as to which domains contact other subunits in the complex) would aid understanding of this figure, and help a reader determine how much of the Ctf13 protein the authors have managed to build.

We have included a more detailed diagram of Ctf13 together with a linear schematic of the protein with assigned secondary structure elements in figure EV2. We have tried to emphasise the uncertainties in the model both in the text and accompanying legend.

Referee #3:

In their study "Structural Basis for Assembly of the CBF3 Kinetochore Complex", Leber and Singleton utilize single particle cryo-electron microscopy (cryo-EM) structural analysis and complimentary biochemical assays to analyze the structure and function of the yeast CBF3 complex. CBF3 is required to recognize the sequence of point centromeres and deposit Cse4 nucleosomes, thereby templating kinetochore assembly. The structure and mechanisms of inner kinetochore proteins remain relatively mysterious, as well as the level of conservation at the level of molecular mechanism despite divergence of kinetochore architecture and specific protein components. This study makes an important contribution, providing significant new mechanistic insight into assembly and regulation of the yeast inner kinetochore.

The authors successfully reconstitute and purify the CBF3 complex, which is composed of a stably-associating "core" of Cep3, Ctf13, and Skp1. Their cryo-EM structure demonstrates that the Ctf13 regulatory subunit, which has not previously been structurally characterized, adopts an FBXL fold which forms a bridging interface between the Cep3 dimer and Skp1. Furthermore, the authors provide a compelling structural rationale for why the complex forms a 1:1:2 Ctf13:Skp1:Cep3 stoichiometry, which is linked to why Cep3 can only engage a single piece of centromeric DNA despite being a dimer. Ctf13 and Skp1 embrace one of the 2 N-terminal Gal4 domains of Cep3, locking it in a stable orientation and sterically occluding its DNA-binding site. Only one such interface can be formed, as a second Ctf13/Skp1 pair would be too large to engage the second Gal4 domain, which remains unobstructed and flexible, and is thus available to bind DNA. Thus, steric occlusion plays a key role in both symmetry breaking during complex assembly and regulation of DNA binding. Finally, the authors find that Skp1 phosphorylation by an unknown kinase regulates DNA engagement by an unresolved mechanism. In sum, this study provides substantial mechanistic insight, and points the way towards an exciting future direction for understanding CBF3 regulation by dissecting the structural and functional effects of Skp1 phosphorylation.

The writing of the manuscript is very clear and lucid. Furthermore, the authors are to be commended for their transparent presentation and interpretation of their structural data. This includes consistent presentation of density maps in their figures and their general interpretation at the appropriate level of detail with atomistic models given the varying resolution, particularly building poorly ordered sections of Ctf13 as poly-alanine. I believe this represents a very nice piece of work, and should be accepted pending minor revisions, detailed below.

Significant points:

Map anisotropy and flexibility of Ctf13

1) The authors note, and the Euler angle density plot presented in Figure S1 supports, 2 preferred orientations of their particles in ice. While the FSC curve suggests an overall resolution of 3.6 Angstroms, it is likely that their resolution is anisotropic, which is apparent in some level of streaking in their density maps, particularly of Ctf13. It is now possible, and would be useful, to quantify this effect utilizing a tool recently released by NRAMM and the Lyumkis lab:

<https://github.com/nysbc/Anisotropy>

If indeed significant anisotropy is present, the authors could try "balancing" the views by excluding a fraction of the particles corresponding to preferred views. This could be implemented in practice by not including particles corresponding to a subset of 2D classes which correspond to these views using the subset selection tool in Relion. As the authors have a very large number of particles, it is possible that this would improve the isotropy of the map without impacting the global resolution.

If this strategy fails to improve the map, it would be useful at a minimum to include the anisotropic resolution assessment in Figure S1.

We have tried to improve the map quality by balancing the number of particles contributing to differing views as suggested, but found it made no difference to the map quality (and the overall FSC was actually worse). We have carried out an analysis of anisotropy as suggested and included the relevant results in Appendix Figure S1D.

2) It would also be worthwhile to attempt to improve the Ctf13 density through focused classification / signal subtraction, as described by Scheres and colleagues:

PMID: 26623517

Since the authors have nice atomic models for Cep3 and Skp1, it should be straightforward to segment their reconstruction using these models (using the "Color zone" / "split map" options in UCSF chimera would be a simple way) to generate a mask defining the region occupied by these molecules. This could then be used in the Relion GUI to subtract the signal corresponding to these subunits from aligned particles, which could then be subjected to 3D classification within an inverse mask corresponding to the region occupied by Ctf13, as described in detail here:

<ftp://ftp.mrc-lmb.cam.ac.uk/pub/scheres/relionreview2016.pdf>

It is possible this procedure will identify discrete conformational states adopted by Ctf13, or one better-ordered structural state, which could then be used to improve the atomic model.

For both points 1 and 2, I would explicitly note that failure to achieve improved maps through these procedures would not impact my overall positive assessment of this study. However, I do think it is important for the authors to make a bona fide attempt to achieve the best possible maps from their data using currently available tools.

We also tried the signal subtraction / focused classification approach as suggested. The approach did lead to some improvement in the density for Ctf13, but we found we were able to get an equal improvement (and superior resolution metrics) by 3D classification of the entire complex. We have updated the methods to make the methodology more explicit. 3D classification did not show any obvious movement of Ctf13 relative to the rest of the complex. We suspect the limited resolution results from local disorder of the secondary structure, and it is worth noting that Ctf13 contains many sections of predicted highly disordered sequence interspersed throughout the LRR domain.

Mechanism of CBF3 regulation via Skp1 phosphorylation

The inhibition of CBF3 by Skp1 phosphorylation is an important observation, and I was surprised that the authors did not make an attempt to rationalize it in the context of their structure, particularly since they honed in on a likely site of phosphorylation, the 37-64 loop.

1) It would be very useful to include a view of the likely phosphorylation loop (a panel in Figure 7 would be appropriate). My impression is that this loop likely faces away from the rest of the structure and thus its effects may be somewhat difficult to rationalize, but this is important information to guide future studies. The implications of the structure for understanding this phosphoregulation should be included in the discussion (at a minimum, what the structure rules out).

We have clarified the text to make it clear that this loop is not actually resolved in the density as mentioned in response to referee 2 (point 2). We have included a panel in figure EV5 (together with the other structural diagrams pertaining to DNA-binding) to show the likely approximate location of the loop, as suggested, and discussed how this may influence phospho-regulation of DNA binding in the discussion.

2) What is the phosphorylation status of the complex the authors utilized for their structural analysis? I'm guessing it was not lambda phosphatase treated? If so, it should be explicitly mentioned that the state visualized likely represents an inhibited state.

The complex was not phosphatase treated for structural studies and we have clarified this in the manuscript.

3) Since they have the material in hand and here do report the preliminary purification of a full CBF3 complex including Ndc10, I believe it would be appropriate to biochemically test and report if lambda phosphatase treatment strengthens the association between the core complex which was visualized and Ndc10 in this paper. If successful, this complex could be the subject of a follow-up study, which is beyond the scope of this manuscript.

We have tried to test this biochemically as suggested. Unfortunately, the rather unstable binding of Ndc10 to the core (in vitro, at least) has made it difficult to get clear data addressing this (and please see response to referee 2, point 1). To partially address the question, we have expressed the full complex with the Skp1 loop deletion construct, presumably mimicking the de-phosphorylated state. We find this construct can pull out increased quantities of Ndc10, suggested that the de-phosphorylation does indeed enhance the interaction (Appendix Figure S3). However, EM analysis of the full complex still suffers from the same issues as previously described (referee 2, point 1). We also note that the core complex including the loop deletion is indistinguishable from the wild-type, at least by negative stain EM (also Appendix Figure S3). We have considered the implications of these findings further in the discussion.

Minor points

1) In Figure 4 C, it is difficult to keep track of which labelled alpha helix corresponds to which protein. Perhaps adding the first letter of that protein (e.g. S or C) would make this clearer. It also appears a label for Skp1 alpha helix 3 has been omitted.

This has been done.

2) In Figure 5 and in the text, the authors claim individual Zinc ions are resolved in their maps, which would be quite surprising at 3.6 Angstroms and is not supported by the density they present in 5b, where separated density peaks for each Zn ion are not apparent. This could be changed to "The C6Zn2 cluster is clearly visualized" in the text and "density for the cluster" in the legend without impacting the interpretation.

We have changed this accordingly.

3) In figure 7D and the legend, the labelling for the mutants is a bit confusing. It is apparent that Mut1 is the actual mutant which produces a lack of competitive binding, while Mut2 is presumably a control which fails to compete. It would be better to label them in a manner which makes this explicit.

The labeling of the mutants has been changed.

4) In Figure S1b, the various FSC curves which are presented are not indicated in the legend. Presumably these correspond to masked, unmasked, and phase-randomized plots generated by Relion. What is what should be explicitly indicated.

The legend has been updated to include the colour code.

5) *The Zheng et al. paper reporting MotionCor2 has been published in Nature Methods: PMID 28250466. This should be cited rather than the BioRxiv preprint.*

The reference has been updated.

Accepted

20 November 2017

Thank you for submitting your revised manuscript for our consideration. It has now been seen once more by two of the original referees (see comments below), and I am happy to inform you that both of them are satisfied with your revisions and strongly recommend publication in The EMBO Journal.

Your article will be processed for publication in The EMBO Journal by EMBO Press and Wiley; below you will find further important information regarding production/publication procedures and license requirements.

REFEREE REPORTS

Referee #1:

This is a very nice manuscript and I congratulate the authors for an important achievement for the kinetochore field. The revised version of the paper is improved and ready for publication.

Referee #3:

In their revision of their study "Structural Basis for Assembly of the CBF3 Kinetochore Complex", Leber, Nans, and Singleton have significantly improved their manuscript. The authors addressed all of my concerns regarding EM data processing; it is clear they have done the best they can with their data. Moreover, the presentation of the results has become substantially clearer, with important details of the structure now being indicated on the figures, as well as discussed more extensively in the text. The expanded discussion, as well the new model Figure 8, is also a big improvement that enhances the accessibility of the study. Finally, the authors made clear the limitations of the "core" complex presented, and why high-resolution structural analysis of the full complex is not feasible.

I recommend immediate acceptance and publication of this study in EMBO Journal.

Corresponding Author Name: Martin Singleton

Manuscript Number: EMBOJ-2017-98134R